# Mining Your Own Secrets: Diffusion Classifier Scores for Continual Personalization of Text-to-Image Diffusion Models

**Saurav Jha**[*]
UNSW Sydney

**Shiqi Yang,**[†] **Masato Ishii, Mengjie Zhao, Christian Simon**
Sony Group Corporation

**M. Jehanzeb Mirza**
MIT CSAIL

**Dong Gong**
UNSW Sydney

**Lina Yao**
CSIRO's Data61 & UNSW Sydney

**Shusuke Takahashi**
Sony Group Corporation

**Yuki Mitsufuji**
Sony Group Corporation & Sony AI

## Abstract

Personalized text-to-image diffusion models have grown popular for their ability to efficiently acquire a new concept from user-defined text descriptions and a few images. However, in the real world, a user may wish to personalize a model on multiple concepts but one at a time, with no access to the data from previous concepts due to storage/privacy concerns. When faced with this continual learning (CL) setup, most personalization methods fail to find a balance between acquiring new concepts and retaining previous ones – a challenge that *continual personalization* (CP) aims to solve. Inspired by the successful CL methods that rely on class-specific information for regularization, we resort to the inherent class-conditioned density estimates, also known as diffusion classifier (DC) scores, for CP of text-to-image diffusion models. Namely, we propose using DC scores for regularizing the parameter-space and function-space of text-to-image diffusion models, to achieve continual personalization. Using several diverse evaluation setups, datasets, and metrics, we show that our proposed regularization-based CP methods outperform the state-of-the-art C-LoRA, and other baselines. Finally, by operating in the replay-free CL setup and on low-rank adapters, our method incurs zero storage and parameter overhead, respectively, over C-LoRA. Project page.

## 1 Introduction

With their photorealistic generation quality and text-guided steerability, text-to-image diffusion models (Saharia et al., 2022; Rombach et al., 2022) have emerged as one of the most flourishing areas in the computer vision community. This has led to their deployment across diverse domains involving the generation of audio/video/3D content, and has, in turn, seen a boost in their commercial value. Despite achieving extraordinary performance, these models typically demand a huge amount of training resources and data. A practical user-centric personalization of these, e.g., using limited data and compute, thus calls for efficient finetuning methods (Kumari et al., 2023; Gal et al., 2023). However, the existing finetuning methods perform poorly on a common real-world scenario, where a model needs to be personalized on sequentially arriving concepts, while being able to generate high-quality images for the previously acquired concepts.

Continual personalization (Smith et al., 2024b) aims to address the above challenge through continual learning (CL) and incorporating of new tasks with unseen concepts while retaining the previously acquired concepts. Naively adapting the existing (non-continual) personalization methods (Kumari et al., 2023; Gal et al., 2023) to acquire a new concept in CL setup often requires a complete re-

---

[*]Work done as an intern at Sony. Correspondence to: `saurav.jha@unsw.edu.au`
[†]Project Lead.

training on data from all the seen concepts that the user desires to generate. However, storing a user's personal data may raise resource/privacy concerns, and can be practically infeasible for edge devices. This calls for a replay-free CL solution, which does away with the seen data once a given concept has been acquired. C-LoRA (Smith et al., 2024b) handles these challenges by learning low-rank adapters (LoRAs) (Hu et al., 2022) per task, where each CL task acquires a single concept. It tackles the forgetting of previous concepts by penalizing the modification of any low-rank matrix spots allocated to these. We find that such a penalty leads to a degenerate CL solution where all LoRA parameter values are encouraged to be near zero (Sec. 3.1). This has further catastrophic consequences for the first task wherein the LoRA parameters are modified in an unconstrained manner and are thus more liable to the penalty. Nevertheless, given their edge in mitigating forgetting (Biderman et al., 2024), we resort to finetuning task-specific LoRAs while still consolidating their previous task knowledge for enabling CL. To the latter end, we note that merging the LoRA parameters based on task arithmetic (Ilharco et al., 2023) remains insufficient at retaining high generation quality. Instead, we propose to exploit the class-specific information of text-to-image diffusion models, with which we can consolidate the model's discriminative semantic knowledge from previous tasks. Our inspiration for this comes from the broader CL literature on classification models where class-specific information, e.g., logits and softargmax scores, are often employed in countering forgetting with regularization (Li & Hoiem, 2017; Buzzega et al., 2020; Jha et al., 2023; 2024).

Namely, we exploit the diffusion classifier (DC) (Li et al., 2023; Clark & Jaini, 2024) scores that encode the semantic concept information inherent to conditional density estimates of pretrained text-to-image diffusion models. We note that leveraging DC scores directly for continual personalization is non-trivial. Instead, we incorporate these into two popular regularization-based CL approaches (Wang et al., 2024): *parameter-space* and *function-space*. For parameter-space regularization, we acknowledge C-LoRA's limitation, and propose Elastic Weight Consolidation (EWC) (Kirkpatrick et al., 2017) in the LoRA parameter space. We then employ DC scores for improving the task-specific Fisher information estimates of EWC for LoRA parameters. For function-space regularization, inspired by deep model consolidation (Zhang et al., 2020), we propose a double-distillation framework that leverages DC scores and noise prediction scores of a diffusion model. Hence, we dub this distillation framework as diffusion scores consolidation (DSC). We consider the practical inefficiencies for computing DC scores in EWC and DSC, and design strategies to overcome these.

We evaluate our proposed consolidation methods qualitatively and quantitatively on four datasets, where the number of images per concept range from 4 to 20. In doing so, we notice the flaw in the existing forgetting metric that quantifies the relative change in previous concept generation quality. Subsequently, we propose to adopt a more robust backward transfer metric that measures the absolute forgetting over tasks. Our experiments on diverse task sequence lengths validate the effectiveness of our methods, all the while requiring zero inference time overhead over C-LoRA. In the spirit of parameter-efficient CL, we explore the compatibility of our method for VeRA (Kopiczko et al., 2024) and multi-concept generation. Lastly, we provide detailed ablations of our design choices with the hope of aiding future personalization works in leveraging DC scores.

## 2 BACKGROUND AND RELATED WORK

**Diffusion models** (Sohl-Dickstein et al., 2015; Ho et al., 2020) are score-based generative models that learn to reverse a gradual noising process. Given an observation $\mathbf{x}_0 \in \mathbb{R}^d$ drawn independently from an underlying data distribution $q(\mathbf{x}_0)$, they approximate $q(\mathbf{x}_0)$ with a variational distribution $p_\theta(\mathbf{x}_0)$, where $\theta$ is the learnable parameter of the diffusion model $\epsilon_\theta$. To achieve this, a forward process corrupts $\mathbf{x}_0$ into increasingly noisy latent variables $\mathbf{x}_1, \ldots, \mathbf{x}_T$ using Gaussian conditional distributions $\prod_{t=1}^{T} q(\mathbf{x}_t|\mathbf{x}_{t-1})$ with a time-dependent variance schedule $\beta_t$. A reverse process then learns $p_\theta$ by starting from $\mathcal{N}(\mathbf{x}_T; \mathbf{0}, \mathbf{I})$ and predicting the gradually decreasing noise at each step (Song & Ermon, 2019; 2020). Although, in general, the shape of the posterior $q(\mathbf{x}_{t-1}|\mathbf{x}_t)$ is unknown, when $\beta_t \to 0$, it converges to a Gaussian (Sohl-Dickstein et al., 2015). Hence, by setting $\alpha_t = 1 - \beta_t$, $q(\mathbf{x}_{t-1}|\mathbf{x}_t)$ can be approximated by modelling the mean $\mu_\theta$ and the variance $\Sigma_\theta$ of $p_\theta$: $p_\theta(\mathbf{x}_{0:T}) = p(\mathbf{x}_T) \prod_{t=1}^{T} p_\theta(\mathbf{x}_{t-1}|\mathbf{x}_t)$, where $p_\theta(\mathbf{x}_{t-1}|\mathbf{x}_t) = \mathcal{N}(\mathbf{x}_{t-1}; \mu_\theta(\mathbf{x}_t, t), \Sigma_\theta(\mathbf{x}_t, t))$. The training objective involves maximizing a variational lower bound on data likelihood, and is achieved by denoising score matching for noise samples $\epsilon \sim \mathcal{N}(0, \mathbf{I})$ and timesteps $t \sim \mathcal{U}[1, T]$:

$$\mathcal{L}_{\text{denoise}} = \mathbb{E}_{\mathbf{x}, \epsilon, \mathbf{c}, t} \left[ \|\epsilon - \epsilon_\theta(\mathbf{x}_t, \mathbf{c}, t)\|_2^2 \right], \tag{1}$$

where $\mathbf{x}_t = \sqrt{\bar{\alpha}_t}\mathbf{x}_0 + \sqrt{1 - \bar{\alpha}_t}\epsilon$, $\bar{\alpha}_t = \prod_{i=1}^{t}\alpha_i$, and $\mathbf{c}$ is the conditioning information (e.g., class). Text-to-image diffusion (Saharia et al., 2022; Rombach et al., 2022) uses a cross-attention mechanism (Vaswani, 2017) in the U-Net (Ronneberger et al., 2015) to guide each reverse process step with a text prompt encompassing $\mathbf{c}$. The keys $\mathbf{K}$ and values $\mathbf{V}$ to the cross-attention are rich semantic embeddings of $\mathbf{c}$ obtained from a pretrained text encoder $\phi$ (like CLIP (Radford et al., 2021a)).

**Personalization** of a text-to-image diffusion model aims to embed a new concept into the model with the goal of generating novel images that incorporate the model's new and prior knowledge. This is achieved by steering the reverse process through a mapping from the textual embedding $\phi(\mathbf{c})$ to the distribution of the latent image features $\mathbf{x}$ (see App. A for further details). DreamBooth Ruiz et al. (2023) and Textual Inversion Gal et al. (2023) perform single-concept personalization by finetuning either all parameters $\theta$ of the diffusion models or by learning a new word vector $V^*$ per new concept. Improving upon these, Custom diffusion (Kumari et al., 2023) performs *parameter-efficient* personalization with the goal of acquiring multiple concepts given only a few examples. They finetune only the weights $\mathbf{W}$ of the key $K$ and value $V$ projection layers in the cross-attention blocks: $\mathbf{W} = [\mathbf{W}^K, \mathbf{W}^V]$, together with regularization on a pretraining prior concept dataset.

**Continual Learning** (CL) (Rolnick et al., 2019; Jha et al., 2020) aims to train a deep neural network on sequentially arriving tasks' data to acquire new knowledge while retaining previously learned knowledge. A popular approach to CL comprises regularization-based methods that mitigate forgetting by imposing a penalty term on the learning objective. Based on how the penalty is computed, regularization may act on the *parameter-space* or the *function-space*. Parameter-space regularization, such as Elastic Weight Consolidation (EWC) (Kirkpatrick et al., 2017), constrains the changes to model weights that were important to previous tasks. EWC uses Fisher information (Fisher, 1922) to measure the parameter importance. Function-space regularization, like Learning without Forgetting (LwF) (Li & Hoiem, 2017) and Deep Model Consolidation (DMC) (Zhang et al., 2020), aims to preserve the model's output behavior on previous tasks. These typically rely on knowledge distillation to ensure that the model's predictions on previous tasks remain consistent. Our work uses EWC and DMC as representatives for the aforesaid regularization techniques.

**Continual Personalization** (Smith et al., 2024b) extends CL to diffusion models for acquiring $N$ sequentially arriving personalization tasks where each task comprises a single user-defined custom concept $n \in \{1, 2, \ldots, N\}$. To respect the real-world privacy and storage concerns, a replay-free CL setup is assumed such that there is no data available from previous tasks. Under such setup, as the number of tasks grow, single-concept adapters stand as poor candidates in terms of resource efficiency and knowledge transferability across tasks. These limitations call for consolidating the $n^{\text{th}}$ task adapter using previous knowledge to enrich it with the nuances of various concepts collectively.

C-LoRA (Smith et al., 2024b) proposes parameter-efficient continual personalization through sequential training of low-rank adapters (LoRA) (Hu et al., 2022) acting on the $n^{th}$ task key and value projection layers $\mathbf{W}_n \in \mathbb{R}^{d_1 \times d_2}$. This allows decomposing $\mathbf{W}_n$ into low-rank residuals: $\mathbf{W}_n = \mathbf{W}_{init}^{K,V} + \sum_{n'=1}^{n-1} \mathbf{A}_{n'}\mathbf{B}_{n'} + \mathbf{A}_n\mathbf{B}_n$, where $\mathbf{A}_n \in \mathbb{R}^{d_1 \times r}$, $\mathbf{B}_n \in \mathbb{R}^{r \times d_2}$, $r$ is the weight matrix rank, and $\mathbf{W}_{\text{init}}^{K,V}$ is the initial pretrained model weight. To tackle forgetting, a self-regularization loss penalizes the $n^{\text{th}}$ task LoRA parameters for altering any previously occupied spot in $\mathbf{W}_n$:

$$\mathcal{L}_{\text{forget}} = \|\left|\sum_{n'=1}^{n-1} \mathbf{A}_{n'}\mathbf{B}_{n'}\right| \odot \mathbf{A}_n\mathbf{B}_n\|^2, \tag{2}$$

where $\|\cdot\|$ is the Frobenius norm, $\odot$ is the element-wise dot product, and $|\cdot|$ is the element-wise absolute value. C-LoRA exploits LoRA for CL to reduce the parameters undergoing interference in incremental training, and to maintain small training/storage overhead (Biderman et al., 2024).

**Classification with diffusion models** (Li et al., 2023; Clark & Jaini, 2024) involves predicting how likely a class $\mathbf{c}_i$ is for an input $\mathbf{x}$ by using a uniform Bayesian prior over all classes $\{\mathbf{c}_1, \mathbf{c}_2, .., \mathbf{c}_N\}$:

$$p_\theta(\mathbf{c}_i \mid \mathbf{x}) = \frac{\exp\{-\mathbb{E}_{\mathbf{x}, \epsilon, \mathbf{c}_i, t}[\|\epsilon - \epsilon_\theta(\mathbf{x}_t, \mathbf{c}_i, t)\|^2]/\tau\}}{\sum_{j=1}^{N}\exp\{-\mathbb{E}_{\mathbf{x}, \epsilon, \mathbf{c}_j, t}[\|\epsilon - \epsilon_\theta(\mathbf{x}_t, \mathbf{c}_j, t)\|^2]/\tau\}}, \tag{3}$$

where $\tau > 0$ is the temperature, and the probabilities $p_\theta = \{p_\theta(\mathbf{c}_1 \mid \mathbf{x}), p_\theta(\mathbf{c}_2 \mid \mathbf{x}), \ldots, p_\theta(\mathbf{c}_n \mid \mathbf{x})\}$ together comprise the Diffusion classifier (DC) scores. The expectation $\mathbb{E}$ is approximated over Monte-Carlo (MC) estimates across (inference) trials. Each trial samples a timestep $t \sim \mathcal{U}[1, T]$,

computes a noisy input $x_t \sim q(\mathbf{x}_t|\mathbf{x}_0)$ and then denoises it using the diffusion model $\epsilon_\theta$ conditioned on the class $\mathbf{c}_i$. DC scores thus help leverage a diffusion model's rich pretrained generation knowledge for classification. Unlike existing works exploiting it only for zero-shot classification, we aim to use DC score as a regularization prior during training such that it can help mitigate forgetting in CL. We note that the computational costs for deriving DC scores are subject to the number of conditional inputs, and the number of trials. To circumvent these, existing works rely on iterative pruning of uninformative classes (Clark & Jaini, 2024), and appropriately choosing the diffusion timesteps across trials (Li et al., 2023). However, these methods still remain practically infeasible during training – iterative pruning per training iteration is computationally intensive while restricting the diffusion timesteps range leads to a loss in the signal reconstruction information. Accordingly, we propose practical considerations for efficient computation of DC scores during training.

## 3  METHOD

In this section, we propose adapting the existing parameter-space and function-space regularization frameworks into our continual personalization setup with LoRA. For each framework, we propose incorporating the class-specific information from DC scores to enrich their regularization. Next, we brief our general CL setup structured to accommodate these frameworks. We then discuss the limitation of C-LoRA that keeps it from being our choice for parameter-space regularization method.

*How do we structure our CL framework for DC scores?* Using DC scores directly while acquiring new concepts can incur significant additional training cost (over single forward pass) given the need for several class-conditional forward passes per training image (Eq. 3). Instead, following Custom Diffusion (Kumari et al., 2023), we learn the $n^{\text{th}}$ concept with a new word vector $V_n^*$ and a LoRA layer by optimizing the diffusion loss (Eq. 1), the prior regularization loss using a common prior concept $\mathbf{c}_0$, and additionally a parameter-regularization loss in case of paremeter-space consolidation. After training, we freeze the word vector, and plug DC scores into two relatively shorter consolidation phases, one for each regularization method. Fig. 2 shows that these phases can work on their own as well as in tandem. Note that we train only one LoRA per task. After consolidation, the $n^{\text{th}}$ task LoRA serves two purposes: (a) handling inference-time queries for $\{1, 2, \ldots, n\}$ tasks, (b) sequentially initializing the $(n + 1)^{\text{th}}$ task LoRA. Next, we detail on each consolidation phase.

### 3.1  DC SCORES FOR PARAMETER-SPACE CONSOLIDATION

**Limitation of C-LoRA.** Despite being a relevant parameter-space consolidation candidate, C-LoRA has been shown to exhibit a loss of plasticity as the self-regularization penalty $\mathcal{L}_{\text{forget}}$ (Eq. 2) increases on longer task sequences (Smith et al., 2024a). Here, we find that $\mathcal{L}_{\text{forget}}$ allows for a more general degeneracy where any learning on new tasks pushes the LoRA weight values toward zero. This not only effects the plasticity but also the stability of C-LoRA, right from the first incremental task ($n = 2$), *i.e.*, when $\mathcal{L}_{\text{forget}}$ first comes into effect. We also find that $\mathcal{L}_{\text{forget}}$ has particularly catastrophic consequences for the first task concept ($n = 1$), where the LoRA weights are learned without any forgetting constraint (see Fig. 1a). This is shown in Fig. 1b, where for task 2, $\mathcal{L}_{\text{forget}}$ decreases throughout training, thus losing most of the information learned for task 1. While imposing a sparsity constraint on the function space of the task 1 LoRA parameters might look plausible at first, we observe that this additional penalty at best delays the degeneracy rather than resolving it (see App. Fig. 7).

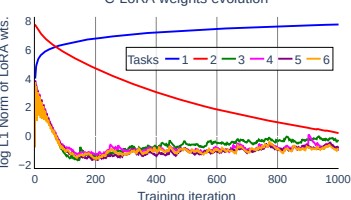

(a) LoRA weights $\mathbf{A}_n\mathbf{B}_n$ for all tasks

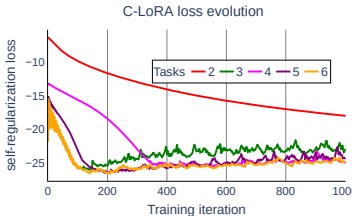

(b) $\mathcal{L}_{\text{forget}}$ for incremental tasks

Figure 1: Task-wise evolution of C-LoRA's: (a) weights, (b) losses.

In light of the above, we instead opt for Elastic Weight Consolidation (EWC) (Kirkpatrick et al., 2017) as our method for parameter-space regularization. While training on $n^{\text{th}}$ task, EWC selectively penalizes the change of parameters $\theta_{n-1} \to \theta_n$ based on their importance to previous tasks. The importance is given by the Fisher Information Matrix (FIM), computed as the expected outer product

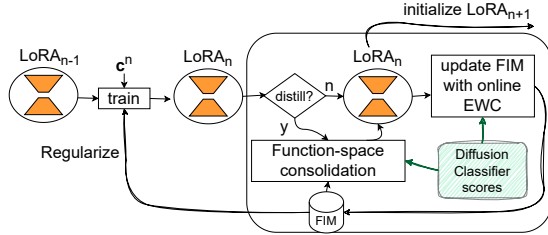

Figure 2: **Framework** for Continual personalization with Diffusion Classifier (DC) scores.

---

**Algorithm 1** Function for computing Denoising and DC scores during consolidation

1: **function** GETDCS($n, \epsilon, \epsilon_\theta, t, \mathbf{x}_t, \mathbf{c}_k$)
   **Input:** $n$: number of seen concepts, $\epsilon$: noise, $\epsilon_\theta$: diffusion model, $t$: timestep, $x_t$: noisy input, $\mathbf{c}_k$: a *subset* of seen concepts to be considered for computing DC scores
   **Output:** $\mathcal{L}_{\text{denoise}}$: $\mathcal{L}_{\text{denoise}}$ scores, $p_\theta$: DC scores, pred: Noise predictions
2:     $\text{pred}[\mathbf{c}_i] = \epsilon_\theta(\mathbf{x}_t, \phi(\mathbf{c}_i), t)$ **for** $\mathbf{c}_i \in \mathbf{c}_k$
3:     $\mathcal{L}_{\text{denoise}}[\mathbf{c}_i] = ||\epsilon - \text{pred}[\mathbf{c}_i]||_2^2$ **for** $\mathbf{c}_i \in \mathbf{c}_k$
4:     $\omega = 1\text{e}^{-10}$                          ▷ Dummy softmax score
5:     $p_\theta = \{\mathbf{c}_i : \omega \text{ for } i \in [0, n]\}$      ▷ DC scores placeholder
6:     $p_\theta[\mathbf{c}_i] = \frac{\exp\{-\mathcal{L}_{\text{denoise}}[\mathbf{c}_i]/\tau\}}{\sum_{j=1}^{|\mathbf{c}_k|} \exp\{-\mathcal{L}_{\text{denoise}}[\mathbf{c}_j]/\tau\}}$ **for** $\mathbf{c}_i \in \mathbf{c}_k$
7:     **return** $\mathcal{L}_{\text{denoise}}, p_\theta, \text{pred}$
8: **end function**

---

**Algorithm 2** One iteration of online EWC

**Input:** $\mathcal{D}_n$: Training data of $n^{\text{th}}$ task, $\epsilon_{\theta_n}$: diffusion model for $n^{\text{th}}$ task, $k$: number of concepts to use for computing DC scores
**Output:** Task-shared Fisher Information Matrix $\mathbf{F}$

1: **for** $\mathbf{x}_0 \in \mathcal{D}_n$ **do**
2:     Sample $t \sim \mathcal{U}([0,1]), \epsilon \sim \mathcal{N}(\mathbf{0}, \mathbf{I})$
3:     $\mathbf{c}_{k-2} = \{\mathbf{c}_i \sim [\mathbf{c}_1, \mathbf{c}_{n-1}]\}$ ▷ sample w/o replacement
4:     $\mathbf{c}_k = \mathbf{c}_{k-2} + \{\mathbf{c}_0, \mathbf{c}_n\}$       ▷ $\mathbf{c}_0 =$ a prior concept
5:     $\mathbf{x}_t = \sqrt{\alpha_t}\mathbf{x}_0 + \sqrt{1 - \alpha_t}\epsilon$
6:     $\mathcal{L}_{\text{denoise}}, p_\theta, - = \text{GETDCS}(n, \epsilon, \epsilon_{\theta_n}, t, \mathbf{x}_t, \mathbf{c}_k)$
7:     $\mathcal{L}_{\text{ewc}} = 0$
8:     **for** $\mathbf{c}_i \in \mathbf{c}_k$:
9:         $\text{gt} = [0, \ldots, 0, i = 1, 0, \ldots, 0]$ ▷ $n + 1$-D vector
10:        $\mathcal{L}_{\text{DC}} = H(p_\theta, \text{gt})$ **if** $i = n$ **else** $- H(p_\theta, \text{gt})$
11:        $\mathcal{L}_{\text{ewc}} += \mathcal{L}_{\text{denoise}}[\mathbf{c}_i] + \delta\mathcal{L}_{\text{DC}}$
12:    Update FIM $\mathbf{F}$ based on $\nabla\mathcal{L}_{\text{ewc}}$
13: **end for**

---

**Algorithm 3** One iteration of DSC

**Input:** $\mathcal{D}_n$: Training data of $n^{\text{th}}$ task, $\{\epsilon_{\theta_1}, \ldots, \epsilon_{\theta_n}\}$: teacher models from $n$ tasks, $k$: number of concepts to use for computing DC scores
**Output:** Student diffusion model $\epsilon_{\theta_s}$ that has been consolidated

1: Initialize: $\epsilon_{\theta_s} = \epsilon_{\theta_n}$, and set teacher-1: $\epsilon_{\theta_n}$
2: **for** $\mathbf{x}_0 \in \mathcal{D}_n$ **do**
3:     Sample $t \sim \mathcal{U}([0,1]), t' \sim \mathcal{U}([0,1])$
4:     Sample $\epsilon \sim \mathcal{N}(\mathbf{0}, \mathbf{I}), \epsilon' \sim \mathcal{N}(\mathbf{0}, \mathbf{I})$
5:     Sample teacher-2 id: $j \sim \mathcal{U}([1, n-1])$
6:     Set teacher-2: $\epsilon_{\theta_j}$
7:     $\mathbf{x}_t = \sqrt{\alpha_t}\mathbf{x}_0 + \sqrt{1 - \alpha_t}\epsilon$                ▷ For $\mathcal{L}_{\text{MSE}}$
8:     $\mathbf{x}_{t'} = \sqrt{\alpha_t}\mathbf{x}_0 + \sqrt{1 - \alpha_t}\epsilon'$               ▷ For $\mathcal{L}_{\text{DC}}$
9:     $\mathbf{c}_k = \{\mathbf{c}_0, \mathbf{c}_j, \mathbf{c}_n\}$               ▷ $\mathbf{c}_0 =$ a prior concept
10:    $-, p_{\theta_b}, \text{pred}_b = \text{GETDCS}(n, \epsilon', \epsilon_{\theta_b}, t, \mathbf{x}_{t'}, \mathbf{c}_k)$ for b in [s,n,j]
11:    $\mathcal{L}_{\text{denoise}} = ||\epsilon - \epsilon_{\theta_s}(\mathbf{x}_t, \phi(\mathbf{c}_n), t)||_2^2$
12:    $\mathcal{L}_{\text{DC}} = H(p_{\theta_n}, p_{\theta_s}) + H(p_{\theta_j}, p_{\theta_s})$
13:    $\mathcal{L}_{\text{MSE}} = ||\text{pred}_s - \text{pred}_n||_2^2 + ||\text{pred}_s - \text{pred}_j||_2^2$
14:    $\mathcal{L} = \mathcal{L}_{\text{denoise}} + \gamma\mathcal{L}_{\text{MSE}} + \lambda\mathcal{L}_{\text{DC}}$
15:    Perform gradient descent on $\nabla\mathcal{L}$
16: **end for**

---

of the gradients of log-likelihood wrt the model parameters:

$$\mathbf{F} \approx \sum_j \nabla_\theta \log p_\theta(\mathbf{c}|\mathbf{x}_n^j)\nabla_\theta \log p_\theta(\mathbf{c}|\mathbf{x}_n^j)^T \approx \sum_j \nabla_\theta \mathcal{L}_{\text{ewc}}^j(\theta)\nabla_\theta \mathcal{L}_{\text{ewc}}^j(\theta)^T, \quad (4)$$

where $\mathbf{c}$ is the class label prediction for an $n^{\text{th}}$ task input $\mathbf{x}_n^j$. The rightmost approximation generalizes the negative log-likelihood to an arbitrary loss function $\mathcal{L}_{\text{ewc}}$. EWC can be viewed as a Laplace approximation to the true Bayesian posterior over the parameters, where the FIM is a proxy for the posterior precision. The choice for the loss function $\mathcal{L}_{\text{ewc}}$ is thus crucial to approximating $\mathbf{F}$. With the goal of improving on this approximation, we incorporate DC scores $p_\theta$ into $\mathcal{L}_{\text{ewc}}$:

$$\mathcal{L}_{\text{ewc}} = \mathcal{L}_{\text{denoise}} + \delta\mathbb{E}_\mathbf{x}\left[\sum_{i=0}^n (2\mathbb{I}_{\{i=n\}} - 1) \cdot H(p_\theta, \mathbf{c}_i)\right], \quad (5)$$

where $\mathbb{I}$ is the indicator function, and $H(a, b) = -a \log b$ is the cross-entropy between the DC scores $p_\theta$ and the class label $\mathbf{c}_i$. The intuition behind Eq. 5 is that for the $n^{\text{th}}$ task images $\mathbf{x}_n$, the DC scores distribution $p_\theta$ should remain closer to the one-hot ground truth for concept $\mathbf{c}_n$, and should be farther from all other class labels $\mathbf{c}_{i<n}$. Fig. 3a and Algo. 2 outline our FIM computation framework.

*How do we adapt EWC to our CL framework?* We emphasize that for each task, we use only one LoRA, which is initialized from the previous task LoRA. While training the $n^{\text{th}}$ task LoRA, we incorporate the EWC regularization term to the loss function. This regularization term uses the FIM computed using the $n - 1^{\text{th}}$ task LoRA. Lastly, we rely on online EWC (Schwarz et al., 2018) to store/update task-shared FIM weights, which are computed using Eq. 5 after training each task.

*How can we compute DC scores efficiently?* Deriving DC scores involves two major computational hurdles: a large number of timestep samples and forward passes for all seen concepts. Previous works using DC scores for test-time classification exploit restricting the timesteps (Li et al., 2023) and iterative pruning of uninformative classes (Clark & Jaini, 2024) as getarounds. However, as we detail below, our training-time setup helps us with tackling these efficiency issues.

**Large number of inference trials:** For the DC scores to converge, the variance of the expectation (Eq. 3) must be low. At test time, this is achieved by averaging the scores accumulated from a

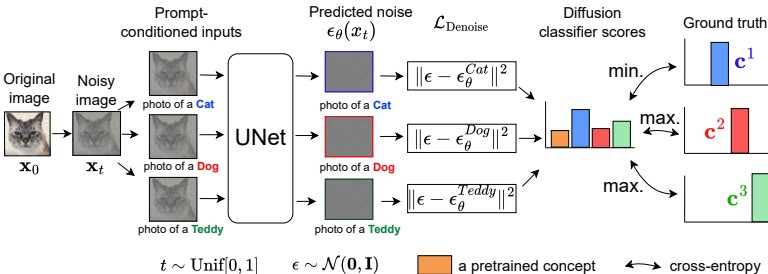

(a) **Derivation of diffusion classifier scores** for FIM computation in parameter-space consolidation.

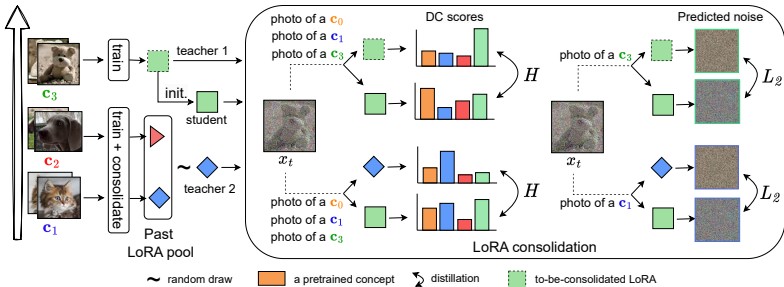

(b) **Illustration of diffusion classifier scores** for function-space consolidation.

Figure 3: **Our consolidation frameworks** for: (a) parameter-space, (b) function-space.

large number ($> 100$) of trials per class. However, during consolidation, we estimate the FIM as an average over multiple epochs (Masana et al., 2023). By using single trial per class per minibatch, our estimated FIM thus incorporates DC scores from diverse range of timesteps over multiple epochs.

**Large number of seen concepts:** Note that the number of class-conditional forward passes for DC score derivation grows linearly with the number of concepts. Here, iteratively pruning the uninformative classes (Clark & Jaini, 2024) still requires multiple passes. Instead, we propose reducing the cost of forward passes to a constant factor using a subset $c_k$ of the number of seen concepts for DC scores computation. This subset always comprises at least two concepts: the task-shared prior concept $c_0$, and the ground truth current task's concept $c_n$. On top of these, we randomly sample $|k-2|$ previous concepts without replacement from the set $\{c_1, ..., c_{n-1}\}$, where $k > 2$ is a hyperparameter chosen by grid search. Note that freezing the word vector $V_i^*$ helps us compute the textual embedding per concept $\phi(c_i)$ once and reuse it throughout the consolidation phase.

## 3.2 DC SCORES FOR FUNCTION-SPACE CONSOLIDATION

As EWC only targets the LoRA parameter values, to fully exploit the information from DC scores, we consider distilling the old LoRA knowledge through function-space consolidation. The intuition behind this (see App. fig. 8) is to guide the diffusion model for generating images that exhibit traits of a conditioned class (Cywiński et al., 2024). For our replay-free CL setup, we use the $n^{\text{th}}$ task images to distill the $n^{\text{th}}$ task LoRA by matching the predictions of a previous task LoRA conditioned on the corresponding previous class of the latter. This involves tackling two *intertwined* CL challenges: (a) alleviating previous concepts' forgetting, and (b) merging the knowledge of old/current LoRAs. To this end, we turn to the Deep Model Consolidation (DMC) framework (Zhang et al., 2020) that uses double distillation to consolidate a student model based on two teachers: the new $n^{\text{th}}$ task model, and the previous $(n-1)^{\text{th}}$ task model (Fig. 3b). Given that our distillation uses diffusion (denoising/DC) scores, we dub our DMC adaptation as Diffusion Scores Consolidation (DSC).

*How do we adapt DMC to our DSC framework?* DMC relies on an external dataset that is chosen to be different from the training data to prevent the consolidation bias towards old or new tasks. However, for large pretrained models like ours, it is practically infeasible to ensure if the external dataset has a distribution that is different from the pre-training data. Moreover, a constant need for external downloads can defeat the purpose of end-to-end CL (Smith et al., 2024b). We thus fall back to the available $n^{\text{th}}$ task training data for DSC. Accordingly, we introduce two *structural* changes

to DMC to improve its effectiveness. First, as we use the current $n^{\text{th}}$ task data, we initialize our student LoRA from the first teacher, *i.e.,* the $n^{\text{th}}$ task LoRA. This makes the consolidation phase a smooth continuation of the $n^{\text{th}}$ task training. Second, given that our learners comprise LoRA rather than heavily parameterized networks, we find that only using the $(n-1)^{\text{th}}$ task LoRA as the second teacher is insufficient for consolidating the student with knowledge of all previous tasks (see App. fig. 9). Instead, we employ all old task LoRA as potential teachers during consolidation. Namely, in each iteration, our second teacher is an old task LoRA sampled at random.

Fig. 3b and Algo. 3 outline the working of DSC. Similar to EWC, we compute DC scores using three concepts: the readily available prior concept $\mathbf{c}_0$, the $n^{\text{th}}$ task concept $\mathbf{c}_n$ learned by the first teacher, and a previous task concept $\mathbf{c}_{j<n}$ learned by the second teacher. The teacher LoRAs should thus generate low-entropy DC scores for their corresponding concepts. The student LoRA is trained to match the DC score distributions from both teachers by minimizing the cross-entropy $H$ for its parameters (Caron et al., 2021). Lastly, while DC scores encode class-specific information for DSC, our primary goal lies in improving the generation quality. We find that relying solely on discriminative DC scores for consolidation is insufficient for noise estimation (see App. fig. 10). Hence, we also incorporate a noise score-matching loss $\mathcal{L}_{\text{MSE}}$ into our DSC learning objective:

$$\mathcal{L}_{\text{DSC}} = \gamma(H(p_{\theta_n}, p_{\theta_s}) + H(p_{\theta_j}, p_{\theta_s})) + \lambda(\mathcal{L}_{\text{MSE}}(\epsilon_{\theta_n}, \epsilon_{\theta_s}) + \mathcal{L}_{\text{MSE}}(\epsilon_{\theta_j}, \epsilon_{\theta_s})), \qquad (6)$$

where $\mathcal{L}_{\text{MSE}}(a, b) = \|a - b\|_2^2$, $p_\theta$ is the DC score, $n$ and $j \sim \mathcal{U}([1, n-1])$ are the first and second teacher task ids, respectively, $s$ is the student id, and $\gamma$ and $\lambda$ are the loss weights. Note that after this consolidation, we discard $\epsilon_{\theta_n}$, and instead use the student $\epsilon_{\theta_s}$ as the $n^{\text{th}}$ task LoRA (see Fig. 2).

*How does DSC differ from existing distillation models?* We note a concurrent branch of distillation methods (Salimans & Ho, 2022; Meng et al., 2023) designed to reduce the number of sampling steps for diffusion model evaluations. While distilling helps us acquire a previous concept using a fraction of the training sample steps, our main purpose behind DSC is to retain the previous task knowledge in a CL setup, and not to optimize on the number of sampling steps for generation. We also note another line of work where distillation is done over a certain timestep range to learn selective features (generic vs domain-specific) from a source diffusion model (Hur et al., 2024). However, we wish to distill the overall knowledge from a teacher, and hence, sample from the entire timestep range $[0, 1]$.

## 4 EXPERIMENTS

**Baselines.** We compare our method with three recent customization methods: Textual Inversion (TI) (Gal et al., 2023), Custom Diffusion (CD) (Kumari et al., 2023), and C-LoRA (Smith et al., 2024b). For CD, we use the best performing variant of Kumari et al. (2023) that trains separate KV parameters per task and then merges them into a single model using a constrained optimization objective; CD EWC uses EWC (Kirkpatrick et al., 2017) with a sequentially trained variant of CD. LoRA sequential trains a LoRA adapter (Hu et al., 2022) for the KV parameters of the CD model in a sequential manner. LoRA merge fuses all the LoRAs with equal weights (Ilharco et al., 2023).

**Implementation.** We use Stable Diffusion v1.4 and v2.0 (Rombach et al., 2022) as backbones based on the Diffusers library (von Platen et al., 2022). Following CD (Kumari et al., 2023), we train all the models for 1000 iterations on all but the Celeb-A setup, where we use 2000 training iterations to capture more fine-grained facial attributes (Smith et al., 2024b). In favor of zero-shot generalization, we finetune our hyperparameters only on the six task sequence of Custom Concept. For both EWC and DSC, we set the number of consolidation iterations to $1/5^{\text{th}}$ of that of the training iterations number. For computing DC scores, the temperature $\tau$ is set to 1.0 for all but the teacher LoRA in DSC where we set $\tau$ to 0.05. The cardinality of $\mathbf{c}_k$ for EWC is set to 5. The DSC loss weights $\gamma$ and $\lambda$ are set to 0.1 and 1.5, respectively. We detail on implementation and hyperparameters in App. H.

**Evaluation.** For each concept, we use DDPM sampling with 50 inference steps to generate 400 images using the prompt "a photo of a $\text{V}_i^*$", where $\text{V}_i^*$ is the modifier token learned for the $i^{\text{th}}$ task (Kumari et al., 2023). For CD and TI, we additionally include the concept name after the modifier token. We encode the generated and the target (real) images using CLIP image encoder (Radford et al., 2021a), and use the features for computing our metrics. We quantify a task's performance using the following metrics computed as the average over all seen concepts: (a) $A_{\text{MMD}}$: the Maximum Mean Discrepancy (MMD) (Gretton et al., 2012) between target and synthetic image feature distribution, (b) CLIP I2I: the CLIP image-alignment (Gal et al., 2023) and (c) KID: the Kernel Inception

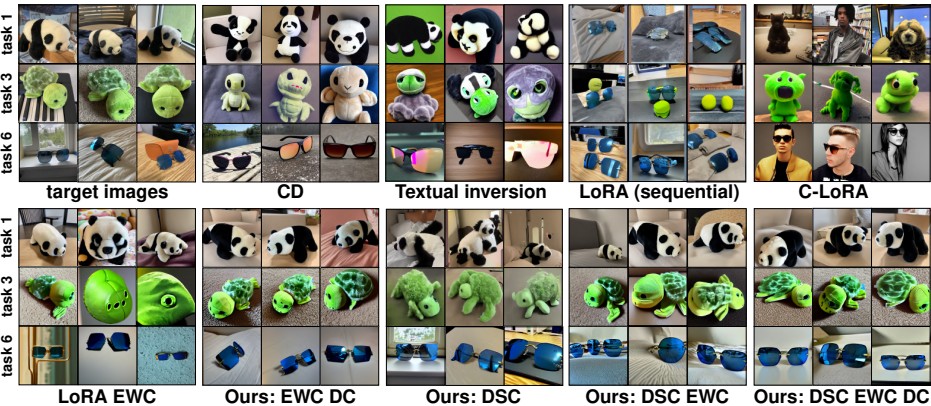

(a) Qualitative results for Custom Concept setup with 6 tasks.

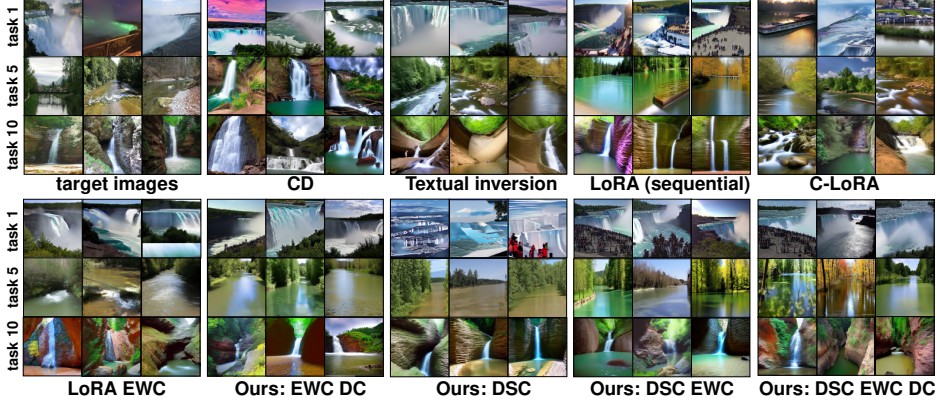

(b) Qualitative results for Landmarks setup with 10 tasks.

Figure 4: **Qualitative results** on CL setups generated after training on all tasks.

Distance (KID) (Bińkowski et al., 2018). To quantify forgetting, we use the forgetting metric $F_{\mathrm{MMD}}$ (Smith et al., 2024b), which is the average change in past task concept generations from task to task. Given the *relative* nature of $F_{\mathrm{MMD}}$, we opt for a more robust backward transfer of MMD: $\mathrm{BwT}_{\mathrm{MMD}}$, which measures the *absolute* change in MMD scores for previous concept generations. We also mention the percentage (wrt the U-Net) of parameters *trained* for a task as $\mathsf{N}_{\mathrm{param}}$ Train, and the percentage of parameters stored between tasks as $\mathsf{N}_{\mathrm{param}}$ Store. We detail these metrics in App. E.

## 4.1 RESULTS

**Continual Personalization of Custom Concepts.** We evaluate our method on the Custom Concept dataset (Kumari et al., 2023) that comprises diverse categories such as plushies, wearables, and toys (see App. D). We randomly sample six such concepts to form six tasks. Figure 4a shows the samples generated for tasks 1, 3, and 6 after training on all 6 tasks: the upper row compares the baselines while the lower row compares our variants. We report the quantitative results in Table 1. Marked by its low $\mathrm{BwT}_{\mathrm{MMD}}$ scores, we find that CD is prone to forgetting the category attributes such as the color, background, and appearance while also struggling with plasticity loss that leads to the generation of incorrect backgrounds for task 6. While Textual Inversion Gal et al. (2023) has zero forgetting due to its frozen backbone, it remains poor at acquiring categories. LoRA (sequential) undergoes catastrophic forgetting of past tasks' concepts (it has the highest $\mathrm{BwT}_{\mathrm{MMD}}$ scores among LoRA-based methods) while C-LoRA displays a higher degree of interference for past tasks. Both these methods have reduced plasticity for acquiring the last task images – sunglasses appear in incorrect numbers/styles. LoRA with EWC shows an improvement on both plasticity and forgetting. Incorporating DC scores into it further helps reduce the forgetting, with an improvement of 3.19 points for $\mathrm{BwT}_{\mathrm{MMD}}$. On the contrary, a naive DSC undergoes huge forgetting as it performs distillation without the previous task data. Using DSC with EWC helps remedy this forgetting, albeit at the cost of reduced plasticity, *e.g.*, task 6 sunglasses with three pairs of glasses. Incorporating DC

Table 1: Custom Concept results at the end of 6 tasks (avg. over 3 seeds)

| Method | $N_{param}$ Train($\downarrow$) | $N_{param}$ Store ($\downarrow$) | KID ($\downarrow$) (x $10^5$) | $A_{MMD}$ ($\downarrow$) (x $10^3$) | BwT$_{MMD}$ ($\uparrow$) | CLIP I2I ($\uparrow$) (x100) | $F_{MMD}$ ($\downarrow$) |
|---|---|---|---|---|---|---|---|
| Textual Inversion | 0.0 | 100.0 | 205.69 | 185.74 | 0 | 60.74 | 0 |
| CD | 2.23 | 100 | 179.4 | 121.89 | -273.41 | 69.53 | 0.62 |
| CD EWC | 2.23 | 101.34 | 177.99 | 121.02 | -245.7 | 69.44 | 0.506 |
| LoRA sequential | 0.09 | 100.0 | 203.11 | 176.38 | -118.56 | 61.30 | 0.052 |
| LoRA merge | 0.09 | 100.54 | 261.74 | 312.44 | -338.06 | 58.29 | 0.043 |
| C-LoRA | 0.09 | 100.54 | 173.8 | 117.2 | -107.47 | 64.89 | 0.034 |
| EWC | 0.09 | 100.54 | 156.91 | 105.07 | -99.34 | 73.19 | 0.008 |
| Ours: EWC DC | 0.09 | 100.54 | 154.25 | 102.81 | -102.53 | **73.41** | **0.0005** |
| Ours: DSC | 0.09 | 100.54 | 187.2 | 198.45 | -105.79 | 73.36 | 0.049 |
| Ours: DSC EWC | 0.09 | 100.54 | 143.92 | 98.0 | -94.63 | 72.92 | 0.02 |
| Ours: DSC EWC DC | 0.09 | 100.54 | **140.18** | **94.1** | **-92.44** | 73.17 | 0.003 |

Table 2: Landmarks results at the end of 10 tasks (avg. over 3 seeds)

| Method | $N_{param}$ Train($\downarrow$) | $N_{param}$ Store ($\downarrow$) | KID ($\downarrow$) (x $10^5$) | $A_{MMD}$ ($\downarrow$) (x $10^3$) | BwT$_{MMD}$ ($\uparrow$) | CLIP I2I ($\uparrow$) (x100) | $F_{MMD}$ ($\downarrow$) |
|---|---|---|---|---|---|---|---|
| Textual Inversion | 0.0 | 100.0 | 107.03 | 95.1 | 0 | 80.26 | 0 |
| CD | 2.23 | 100 | 376.3 | 249.85 | -183.02 | 56.95 | 0.047 |
| CD EWC | 2.23 | 102.23 | 377.76 | 250.79 | -191.2 | 56.88 | 0.007 |
| LoRA sequential | 0.09 | 100.0 | 169.83 | 92.55 | -112.79 | 74.6 | 0.063 |
| LoRA merge | 0.09 | 100.9 | 225.11 | 144.69 | -173.5 | 67.81 | 0.009 |
| C-LoRA | 0.09 | 100.9 | 104.63 | 68.77 | -16.47 | 78.06 | 0.014 |
| EWC | 0.09 | 100.9 | 66.76 | 44.53 | -43.06 | 81.84 | 0.008 |
| Ours: EWC DC | 0.09 | 100.9 | 56.98 | 38.43 | -5.57 | **82.86** | 0.035 |
| Ours: DSC | 0.09 | 100.9 | 111.69 | 73.2 | -72.07 | 77.44 | 0.056 |
| Ours: DSC EWC | 0.09 | 100.9 | 87.57 | 57.75 | -5.91 | 80.15 | **0.004** |
| Ours: DSC EWC DC | 0.09 | 100.9 | **51.22** | **36.09** | **-3.85** | 82.19 | 0.043 |

scores with DSC EWC further helps improve on these results, and is our best performing variant as it displays the least KID, $A_{MMD}$, and BwT$_{MMD}$ scores (see App. F for taskwise performances).

**Continual Personalization of Landmarks.** We study the compared methods on natural landmarks, using a 10 task sequence from the Google Landmarks dataset v2 (Weyand et al., 2020). We create tasks by sampling *global* waterfall landmarks with 20 images each (see App. D). This is a challenging setup given the distinct geographical features of the waterfall landmarks despite them being overall similar. Fig. 4b shows the qualitative results from tasks 1, 5, and 10 generated after training on all 10 tasks. Table 2 reports the quantitative results. Here, CD displays catastrophic forgetting of the middle task 5. LoRA sequential shows reduced plasticity for acquiring task 10, as also marked by its high KID and $A_{MMD}$ scores. Similar to Custom Concept, C-LoRA displays a high degree of forgetting for task 1 while LoRA EWC helps remedy this to some extent. Our DSC-only variant remembers the generic previous task traits but struggles to accurately generate their finer details, e.g. multiple waterfalls for task 1. Using EWC with DSC helps resolve this, and plugging in the DC scores further helps improve on the results. Namely, our DSC EWC DC variant performs the best on 3 out of 5 performance metrics including the robust backward transfer score. Finally, we note that our proposed variants have zero parameter overhead over C-LoRA for training and storage.

**Continual Personalization of Household objects and Real Faces.** We further compare the LoRA-based methods on finegrained images of household objects from the Textual Inversion (TI) dataset (Gal et al., 2023), and that of celebrity faces from the CelebA dataset (Liu et al., 2015). These setups comprise task sequences of length 9 and 10, respectively (see App. D for details). We leave the qualitative and quantitative results in App. F.2. We find the results to follow the same pattern as with Custom Concept and Landmarks. For both these setups, C-LoRA fails to remember the finegrained details of previous concepts, e.g., exact object/facial attributes, and instead, only retains the overall feature, e.g., the dominant color. Also, our DSC EWC DC variant performs the best here.

**Long task sequence.** We study the scalability of our proposed methods to a sequence of 50 concepts chosen at random from the Custom Concept dataset (Kumari et al., 2023), with variable number of training images per concept. To avoid any learning bias from large early tasks with more training images, we pick all 50 tasks at random rather than adding them over our six tasks sequence. App. fig. 13 compares the results of C-LoRA, LoRA EWC, and Ours (EWC DC). Similar to Smith et al. (2024b), we find that the performance of C-LoRA saturates as the number of tasks grow. Instead, applying EWC on the LoRA parameters emerges as a better performer on the long run. EWC with DC retains the performance particularly on the latter ($> 35$) tasks (see App. F.1 for further results).

## 4.2 ABLATION STUDIES

We ablate the influence of DC scores in our training objective. We list two sanity checks to ensure that our framework leverages DC scores. We discuss the impact of the number of concepts used for computing DC scores, and leave further hyperparameter ablations in App. H.

**Sanity check I: DC scores reduce the uncertainty in FIM estimation.** We perform top-5 Eigenvalue analysis for the FIM computed with and without DC scores. Fig. 5 shows that DC scores helps capture larger eigenvalues for the same LoRA parameter. Intuitively, this means that the parameters are more strongly informed by the data regarding the directions of high likelihood changes. We leave the analyses of further layers in App. fig. 17.

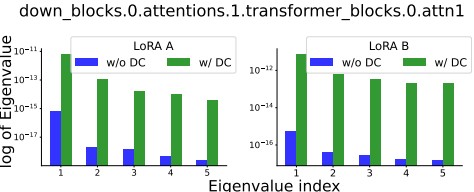

Figure 5: Top-k eigenvalue analysis for FIM.

**Sanity check II: DC scores help with training set classification.** For DC scores to help enhance the generative quality of tasks, their classification information needs to be reliable, *i.e.,* consolidation with wrong classification scores should interfere with the generation results. To validate this, we probe the classification accuracy of different methods on training data of incremental tasks, after training on all six tasks of Custom Concept. As shown in Fig. 6, consolidating with DC scores endows us with classification gains on the overall training data.

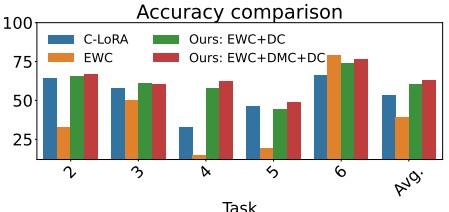

Figure 6: Training set accuracy of different methods on the Custom Concept setup.

**Impact of the number of concepts $k$ for DC scores computation.** We study how the number of randomly sampled previous task concepts effects the performance of EWC DC (see App. fig. 15). We notice that excluding the common prior concept $\mathbf{c}_0$ in DC scores computation, *i.e.,* $k = 2$, leads to the worst performance overall. All setups with $k > 2$ perform similar until task 2 as there is only one available previous concept. From task 3 onward, $k = 3$ still gets to sample only one previous concept per iteration while $k = 7$ can use all previous concepts at each step. We find that the performance of EWC DC saturates as $k$ increases beyond 5. This is because not all previous concepts carry useful discriminative information for reliable DC scores. We use $k = 5$ throughout.

**Training time complexity.** App. table 9 shows that our proposed consolidation methods scale *linearly* in the training sample size whereas C-LoRA scales *bilinearly* in the training sample size and the number of tasks. This implies that while on shorter task sequences, the average runtime per training iteration of our methods remains higher than C-LoRA (5.3s for EWC, 5.7s for DSC, 0.8s for C-LoRA on 6 tasks) given the several conditional forward passes needed for DC scores computation, the time gap per iteration between C-LoRA and our methods bridges as the number of tasks grow: 5.5s for EWC, 5.72s for DSC, 3.8s for C-LoRA on 50 tasks (see App. I for discussion).

**Compatibility for VeRA and Multi-task generation.** In App. Sec. G.2- G.3, we show that our proposed method is compatible with VeRA (Kopiczko et al., 2024) (where we replace LoRA with VeRA, and train as usual) and with multi-concept generation (where prompts include two concepts).

## 5 CONCLUSION

In this paper, we propose continual personalization of pretrained text-to-image diffusion models using their inherent class-conditional density estimates, *i.e.,* Diffusion classifier (DC) scores. Namely, we alleviate forgetting using DC scores as regularizers for parameter-space and function-space consolidation. We show the superior performance of our methods through extensive quantitative and qualitative analyses across diverse CL task lengths. While DC scores have previously been utilized for test-time classification (Clark & Jaini, 2024), we are the first to advocate that such pre-trained class-specific knowledge of diffusion models can further be reinforced through fine-tuning and can help enhance the performance of downstream generation tasks. We thus hope that our work paves the general way for leveraging DC scores in personalization of pretrained conditional diffusion models.

## 6 ACKNOWLEDGEMENT

We are grateful to Yuhta Takida for their feedback and comments. We would like to acknowledge Sander Dieleman's blog (Dieleman, 2024) for insights on diffusion distillation.

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

---

**Algorithm 4** Algorithm summarizing our training and consolidation workflow.

---

**Input:** $\mathcal{D}_n$: Training data of $n^{\text{th}}$ task, $\{\epsilon_{\theta_1}, \ldots, \epsilon_{\theta_{n-1}}\}$: teacher models from previous $n-1$ personalization tasks, $k$: the number of concepts to use for computing DC scores, $\mathbf{F}$ : the task-shared Fisher Information Matrix (FIM) encoding the Fisher information from $n-1$ tasks
**Output:** the diffusion model $\epsilon_{\theta_n}$ consolidated with the knowledge of all $n$ personalization tasks, the task-shared FIM $\mathbf{F}$ updated with the parameter importance for the $n^{\text{th}}$ task

1: $\epsilon_{\theta_n} \leftarrow \epsilon_{\theta_{n-1}}$             ▷ Sequential initialization
2: $\epsilon_{\theta_n} \leftarrow$ Train $\epsilon_{\theta_n}$ on $\mathcal{D}_n$ using the standard denoising score matching objective (Eq. 1) and the Fisher penalty (Kirkpatrick et al., 2017) based on $\mathbf{F}$      ▷ Parameter-space consolidation
3: $\epsilon_{\theta_s} \leftarrow$ Perform DSC based on Algo. 3         ▷ Function-space consolidation
4: $\epsilon_{\theta_n} \leftarrow \epsilon_{\theta_s}$        ▷ Replace the $n^{\text{th}}$ task LoRA with the consolidated LoRA
5: $\mathbf{F} \leftarrow$ Perform online EWC based on Algo. 2    ▷ Update $\mathbf{F}$ with Fisher information for $n^{\text{th}}$ task
6: **return** $\epsilon_{\theta_n}, \mathbf{F}$

---

# A    PERSONALIZATION IN TEXT-TO-IMAGE DIFFUSION MODELS

**Personalization** of a text-to-image diffusion model aims to embed a new concept into the model by steering the reverse process through a mapping from the textual embedding $\phi(\mathbf{c})$ to the distribution of the latent image features $\mathbf{x}$, where $\phi$ is the text encoder. To do so, the text-to-image cross-attention blocks in the U-Net consider the query $\mathbf{Q} = \mathbf{W}^Q \mathbf{x}$, the key $\mathbf{K} = \mathbf{W}^K \phi(\mathbf{c})$, the value $\mathbf{V} = \mathbf{W}^V \phi(\mathbf{c})$, and perform the weighted sum operation: $\mathsf{softmax}\left(\frac{\mathbf{Q}\mathbf{K}^T}{\sqrt{d'}}\right)\mathbf{V}$, where the weights $\mathbf{W}^Q$, $\mathbf{W}^K$, and $\mathbf{W}^V$ map the input $\mathbf{x}$ and $\mathbf{c}$ to $\mathbf{Q}, \mathbf{K}$, and $\mathbf{V}$, respectively, and $d'$ is the output dimension. Custom diffusion (Kumari et al., 2023) perform *parameter-efficient* personalization with the goal of acquiring multiple concepts given only a few examples. They show that upon finetuning on a new concept, the text-projection weights $\mathbf{W}^K, \mathbf{W}^V$ of the text-to-image cross-attention blocks in the U-Net undergo the highest rate of changes. Subsequently, they finetune only the cross-attention weights $\mathbf{W} = [\mathbf{W}^K, \mathbf{W}^V]$ together with regularization, rare token embedding initialization, and constrained weight merging. C-LoRA builds upon this parameter-efficient setup and further proposes training low rank adaptrs (LoRA) (Hu et al., 2022) for the cross-attention layers in the U-Net. Subsequently, we consider using LoRA as well.

*How do we obtain and leverage the new word vector $V_n^*$ in the training process?* Following TI (Gal et al., 2023) and CD (Kumari et al., 2023), to personalize our text-to-image diffusion model on a new concept $\mathbf{c}_n$, we introduce a new token representing this concept and learn its corresponding word vector $V_n^*$ by optimizing only this embedding and the $n^{\text{th}}$ task LoRA while keeping the rest of the model's parameters frozen. To do so, we create prompts for the $n^{\text{th}}$ concept that include the new token (e.g.,"a photo of $[V_n^*]$"). By inputting these prompts into the model and comparing the generated images with the training images, our standard denoising score matching objective (Eq. 1) measures how well the model reproduces the concept. Minimizing this loss adjusts the word vector $V_n^*$ so that the model associates the new token with the visual characteristics of $\mathbf{c}_n$, enabling it to generate images of the concept when the token is used in prompts. Lastly, as mentioned in Sec. 3, $V_n^*$ is acquired during the training stage and remains frozen thereafter, *i.e.,* while we perform consolidation using the DC scores.

## B  C-LoRA with sparsity constraint on Task-1 LoRA parameters

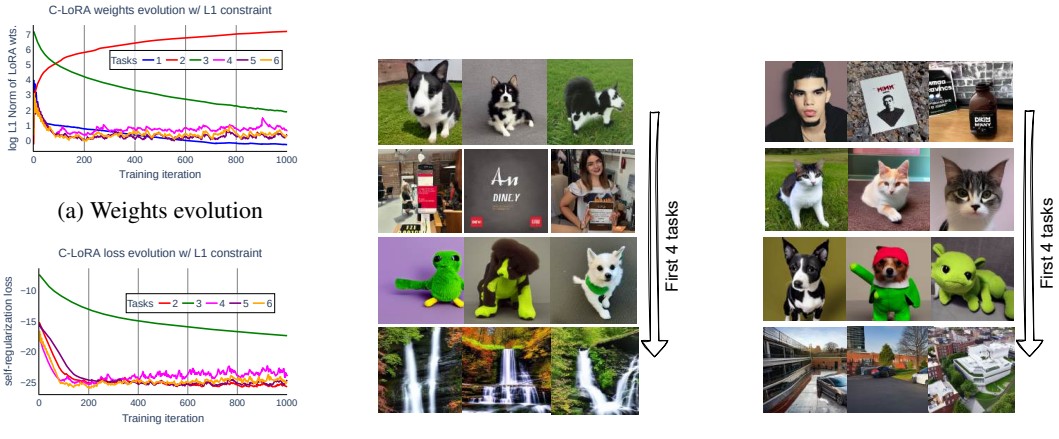

(a) Weights evolution

(b) Loss evolution

(c) C-LoRA results with L1-norm on first task.

(d) C-LoRA results (without the L1-norm constraint).

Figure 7: **C-LoRA with sparsity constraint on the first task:** To overcome the catastrophic forgetting of first task in C-LoRA (Smith et al., 2024b) (see Sec. 3.1), we consider restricting LoRA weight updates during first task by incorporating a sparsity (L1 norm) constraint into the training objective for the first task. However, as shown in Fig. 7a and 7b, this merely results in the degenerate solution shifted by one task, *i.e.,* now the task 2 weights (instead of task 1) undergo significant updates, which in turn causes $\mathcal{L}_{\text{forget}}$ for task 3 to decrease throughout training (as most of the task 2 spots get edited). Fig. 7c shows the results generated by this model for the first four tasks on our Custom Concept setup. Compared to the results of C-LoRA in Fig. 7d, now even task 2 image generation (for the pet cat concept) is seen to exhibit catastrophic forgetting.

## C  Diffusion Scores Consolidation (DSC) for Function-space Regularization

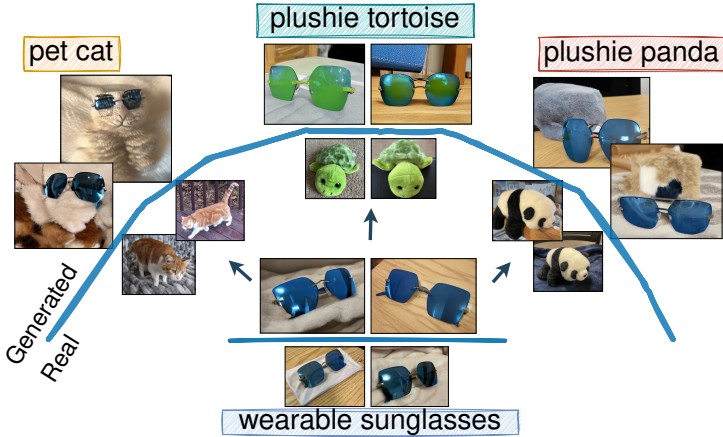

Figure 8: **Motivation behind function-space consolidation:** conditioning current task images (wearable sunglasses) on previous classes (those around the circumference) helps generate images that share features with the previous classes (Cywiński et al., 2024). In the absence of replay samples (real images) from previous personalization tasks, we exploit the aforesaid property using the current ($n^{\text{th}}$) task images to distill the current task LoRA (finetuned on wearable sunglasses) by matching the predictions of the LoRA corresponding to the previous tasks on their respective previous task concepts. Images have been resized to highlight the subject of interest. The real images for previous concepts have been provided for the sake of reference.

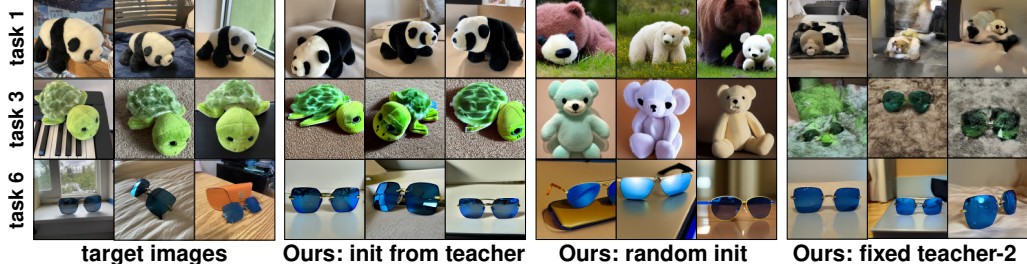

Figure 9: **Design choices and their results for our DSC framework** (from left to right): (A) the ground truth target images for tasks 1, 3, and 6 of our Custom Concept CL setup; (B) generated results for our proposed DSC EWC DC framework, as also reported in Fig. 4a; (C) generated results for the DSC EWC DC framework where we follow DMC (Zhang et al., 2020) to initialize our student LoRA using random weights: the consolidated student fails to properly acquire the previous and current task custom categories; (D) generated results for the DSC EWC DC framework where we follow DMC (Zhang et al., 2020) to use the $(n-1)^{th}$ task LoRA as our fixed second teacher, rather than randomly sampling the second teacher from the pool of all previous task LoRAs: the consolidated student undergoes catastrophic forgetting of previous concepts.

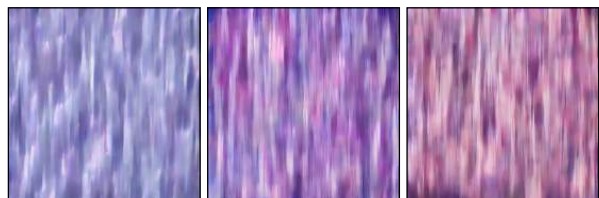

Figure 10: **LoRA consolidated using only DC scores in DSC** (after training on task 2) generates unintelligible images. We thus opt for using DC scores together with Denoising score matching in our DSC framework (Eq. 6).

## D  DATASETS AND THEIR CONCEPTS

Four our **Custom Concept** setup, we select the following 6 classes from the CustomConcept101 dataset (Kumari et al., 2023) with at least nine images each: furniture sofa1, plushie panda, plushie tortoise, garden, transport car 1, and wearable sunglasses 1.

For our **Google Landmarks v2** (Weyand et al., 2020) setup, we select 10 such geographically diverse waterfall landmarks and download 20 images for each. These landmarks (and their countries) include: Bow falls (Canada), Davis falls (Nepal), Fukuroda falls (Japan), Huka falls (New Zealand), Iguazu falls (Argentina), Korbu falls (Russia), Mang falls (China), Niagara falls (Canada), Rufabgo falls (Russia), and Shin falls (the UK).

Our **Textual Inversion** dataset (Gal et al., 2023) setup simply uses all nine available household categories and their images: cat statue, clock, colorful teapot, elephant, mug skulls, physics mug, red teapot, round bird, and thin bird. Note that some of these concepts contain as few as four images.

For our **Celeb-A** (Liu et al., 2015) setup, we rely on its $256 \times 256$ resized version from Kaggle.[1] We choose 10 such celebrities at random that have at least 15 images. Their IDs include: 2079, 3272, 4407, 4905, 5239, 5512, 5805, 7779, 8692, 9295.

---

[1] https://www.kaggle.com/datasets/badasstechie/celebahq-resized-256x256

## E   METRICS DEFINITION

Following Smith et al. (2024b), we report (i) $N_{param}$ Train as the percentage of parameters (with respect to the U-Net backbone) that are trainable while learning a task and (ii) $N_{param}$ Store as the percentage of parameters that are stored over the entire task sequence. Let $N$ be the number of personalization tasks, where each task $j \in \{1, 2, \ldots, N\}$ comprises a dataset $\mathcal{D}$ hosting a single personal concept. Let $X_{i,j}$ be the generated images for the $j^{\text{th}}$ task by a model trained sequentially until the $i^{\text{th}}$ task, and $X_{\mathcal{D},j}$ be the corresponding original dataset images for the $j^{\text{th}}$ task. Then, using a pretrained CLIP model $\mathcal{F}_{clip}$ (Radford et al., 2021b) as the feature extractor, we define (iii) the average of the maximum mean discrepancy (MMD) $A_{\text{MMD}}$ metric (where lower is better) over $N$ tasks as:

$$A_{\text{MMD}} = \frac{1}{N} \sum_{j=1}^{N} \text{MMD}(\mathcal{F}_{clip}(X_{\mathcal{D},j}), \mathcal{F}_{clip}(X_{N,j})) \tag{7}$$

where the MMD is computed using a quadratic kernel function (Gretton et al., 2012). Accordingly, (iv) the forgetting metric $F_{\text{MMD}}$ (where lower is better) quantifies how much the generated images have diverged due to sequential training:

$$F_{\text{MMD}} = \frac{1}{N-1} \sum_{j=1}^{N-1} \text{MMD}(\mathcal{F}_{clip}(X_{j,j}), \mathcal{F}_{clip}(X_{N,j})) \tag{8}$$

As also stated in Sec. 4, the forgetting metric $F_{\text{MMD}}$ is a *relative* measure of divergence with respect to the results generated by the $j^{\text{th}}$ task model. It can thus be misleading for cases where the $j^{\text{th}}$ task model is itself a poor learner but does not forget much (possibly because it allocates only a tiny fraction of the parameter space for learning new tasks). To the end goal of quantifying forgetting more reliably, we introduce (v) the backward transfer metric $\text{BwT}_{\text{MMD}}$ that measures how much the generated images have diverged from their *absolute* ground truth counterparts as a result of sequential training:

$$\text{BwT}_{\text{MMD}} = \frac{1}{N-1} \sum_{j=1}^{N-1} \Big( \text{MMD}\big(\mathcal{F}_{clip}(X_{\mathcal{D},j}), \mathcal{F}_{clip}(X_{j,j})\big) - \text{MMD}\big(\mathcal{F}_{clip}(X_{\mathcal{D},j}), \mathcal{F}_{clip}(X_{N,j})\big) \Big) \tag{9}$$

Contrary to $F_{\text{MMD}}$, a larger $\text{BwT}_{\text{MMD}}$ is desirable as it implies that learning the $N^{th}$ task helps improve the generative quality of the $j^{th}$ task images. Following Kumari et al. (2023), we additionally include the following metrics that leverage the pretrained CLIP model's features: (vi) the *image alignment* quantifying the visual similarity of the generated images with their ground truth targets in the CLIP visual feature space, (vii) the *text alignment* quantifying the text-to-image similarity of the generated images with their respective prompts in the CLIP multimodal feature space, and (viii) the *Kernel Inception Distance* (KID) Bińkowski et al. (2018) quantifying overfitting on the target concept (e.g., $V^*$ panda) due to the forgetting of the pretrained knowledge (e.g., panda).

In terms of magnitude, **the higher the better** ($\uparrow$) holds for: the image alignment (I2I) and the backward transfer ($\text{BwT}_{\text{MMD}}$) metrics, while **the lower the better** ($\downarrow$) holds for all other metrics.

## F   MAIN RESULTS (CONTINUED)

We study the taskwise performance ($A_{\text{MMD}}$ and KID) evolution of the compared methods on our CL setups of Custom Concept and Landmarks. Fig. 11a and 11b show that overall, our DC score-based variants perform better against their non-DC score-based counterparts as well as against other baselines on every incremental task.

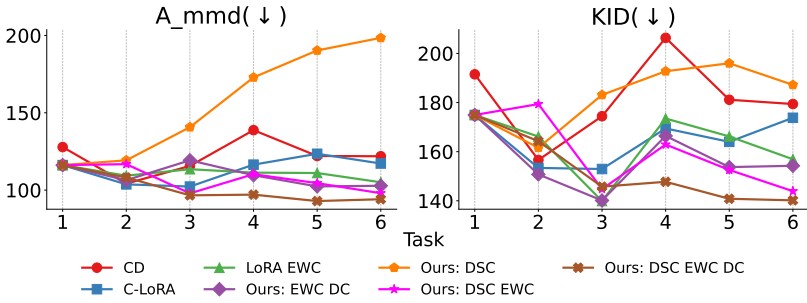

(a) Taskwise performance evolution on Custom Concept CL setup.

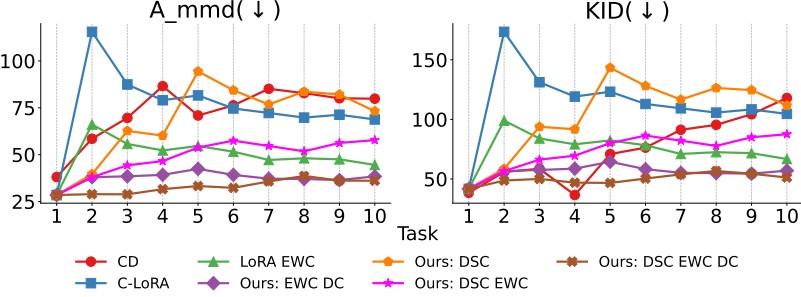

(b) Taskwise performance evolution on Landmarks CL setup.

Figure 11: Taskwise performances.

## F.1 RESULTS FOR LONG TASK SEQUENCE

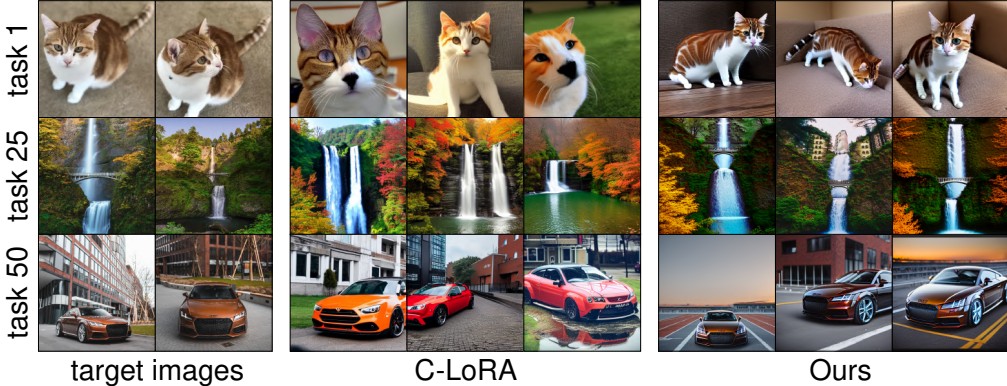

Figure 12: **Qualitative results for 50 tasks Custom Concept setup:** we compare the results of our EWC DC variant with that of C-LoRA for generating the images from tasks 1, 25, and 50. We find that C-LoRA does not only suffer from a loss of plasticity to acquire the $50^{th}$ task images but has also undergone catastrophic forgetting of the first task images, which is inline with our findings from Sec. 3.1. On the contrary, our method scales well to mitigate the forgetting of the previous tasks while still remaining plastic enough to acquire the $50^{th}$ task.

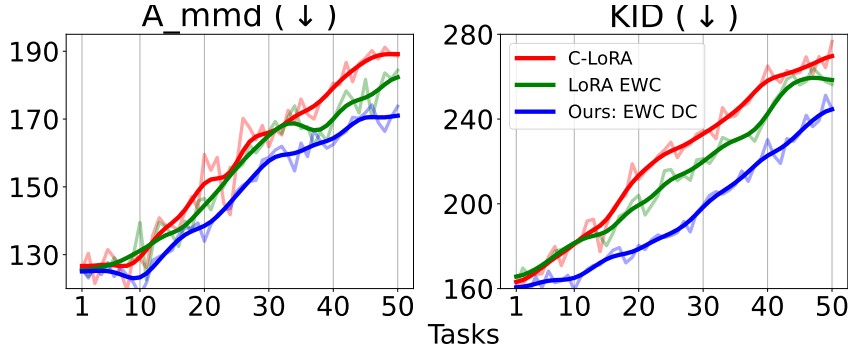

Figure 13: **Performance evolution on a sequence of 50 custom concepts** (Kumari et al., 2023). Shaded lines represent absolute values while solid lines denote the simple moving averages over tasks.

## F.2 RESULTS ON HOUSEHOLD OBJECTS AND REAL FACES

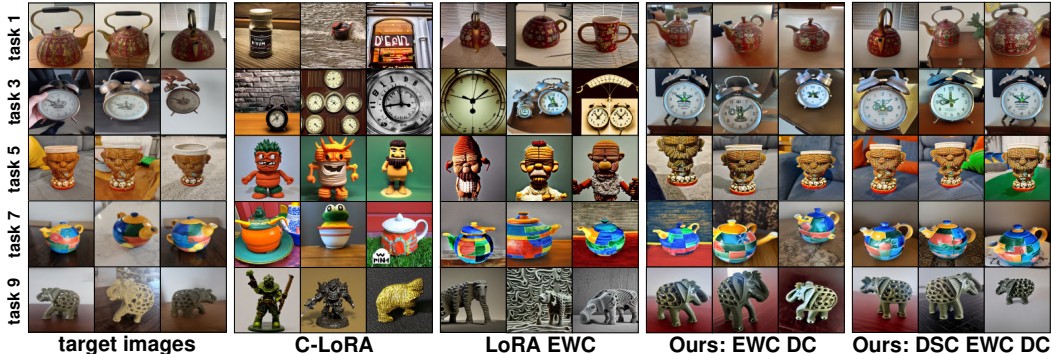

Figure 14: **Qualitative results** for LoRA-based methods on Textual Inversion dataset (Gal et al., 2023) setup with 9 tasks.

Table 3: TI results after 9 tasks (avg. over 3 seeds): (↓) indicates lower is better.

| Method | $N_{param}$ Train(↓) | $N_{param}$ Store (↓) | KID (↓) (x $10^5$) | $A_{MMD}$ (↓) (x $10^3$) | BwT$_{MMD}$ (↑) | CLIP I2I (↑) (x100) | $F_{MMD}$ (↓) |
|---|---|---|---|---|---|---|---|
| CD | 2.23 | 100.0 | 296.11 | 187.4 | -204.31 | 68.13 | 0.025 |
| LoRA sequential | 0.09 | 100.0 | 261.07 | 132.58 | -125.03 | 67.1 | 0.008 |
| C-LoRA | 0.09 | 100.81 | 181.33 | 105.7 | -106.16 | 69.04 | 0.047 |
| EWC | 0.09 | 100.81 | 158.93 | 88.21 | -93.18 | 71.46 | **0.003** |
| Ours: EWC DC | 0.09 | 100.81 | 139.66 | 82.75 | -79.0 | **73.08** | 0.047 |
| Ours: DSC | 0.09 | 100.81 | 168.09 | 96.41 | -135.74 | 69.81 | 0.008 |
| Ours: DSC EWC | 0.09 | 100.81 | 135.61 | 80.6 | -84.11 | 71.08 | 0.008 |
| Ours: DSC EWC DC | 0.09 | 100.81 | **121.08** | **78.34** | **-73.18** | 72. 66 | 0.005 |

## G ADDITIONAL RESULTS

### G.1 COMPATIBILITY WITH STABLE DIFFUSION V2.0

For a fair comparison with the state-of-the-art (Smith et al., 2024b), we report our main results using Stable Diffusion v1.4. We further note that Stable Diffusion v1.4 and v1.5 remain architecturally

Table 4: Celeb-A results after 10 tasks (avg. over 3 seeds): ($\downarrow$) indicates lower is better.

| Method | $N_{param}$ Train($\downarrow$) | $N_{param}$ Store ($\downarrow$) | KID ($\downarrow$) (x $10^5$) | $A_{MMD}$ ($\downarrow$) (x $10^3$) | BwT$_{MMD}$ ($\uparrow$) | CLIP I2I ($\uparrow$) (x100) | $F_{MMD}$ ($\downarrow$) |
|---|---|---|---|---|---|---|---|
| CD | 2.23 | 100.0 | 317.0 | 219.82 | -215.14 | 42.9 | 0.006 |
| LoRA sequential | 0.09 | 100.0 | 286.55 | 308.2 | -156.22 | 60.79 | 0.061 |
| C-LoRA | 0.09 | 100.9 | 151.51 | 101.01 | -101.97 | 64.38 | 0.012 |
| EWC | 0.09 | 100.9 | 141.16 | 95.07 | -92.78 | 66.37 | 0.01 |
| Ours: EWC DC | 0.09 | 100.9 | 127.0 | 87.06 | -86.43 | 67.34 | 0.004 |
| Ours: DSC | 0.09 | 100.9 | 185.62 | 112.44 | -97.5 | 66.81 | **0.003** |
| Ours: DSC EWC | 0.09 | 100.9 | 138.69 | 92.49 | -94.13 | 66.29 | 0.061 |
| Ours: DSC EWC DC | 0.09 | 100.9 | **119.5** | **91.72** | **-82.71** | **68.3** | 0.004 |

Figure 15: **Effect of the number of concepts** $k$ for DC scores computation on the six-tasks Custom Concept setup performance (refer to Algo. 2 on the usage of $k$). *Interpretation:* $k$ dictates the cardinality of the set $c_k$ that we leverage for preventing the cost of DC scores computation to grow with the number of seen concepts. $c_k$ usually comprises the current $n^{th}$ task concept $c_n$ and the readily available common prior concept $c_0$. On top of these, we randomly sample $k - 2$ previous task concepts that are to be employed in the DC scores computation. As such, all setups with $k > 2$ perform similar until task 2 as there is only one available previous concept in the pool to sample from. From task 3 onward, $k = 3$ still gets to sample only one previous concept per iteration while $k = 5$ can use all previous concepts per iteration (note that our Custom Concept setup has a total of six concepts, and thus the maximum number of previous task concepts is 5 which is during training on the sixth task). We find that the performance of EWC DC saturates as $k$ increases beyond 5. This is because not all previous concepts carry useful discriminative information for deriving reliable DC scores. We thus use $k = 5$ across all our setups. Lastly, note that DC scores comprise probability distribution over concepts and thus require the minimum number of 2 concepts for derivation. As a result, we include the pre-trained common prior concept $c_0$ besides the current $n^{th}$ task concept $c_n$ into $c_k$ on all but the $k = 2$ setting. On the latter setting, we sample 1 previous concept at random per iteration, which is similar to the $k = 3$ setup except for the common prior concept included in $c_k$. We find that excluding the common prior concept leads to the worst performance overall. This could be because the prior concept images might help preserve more discriminative semantic information in the DC scores when compared to other downstream task concepts.

very similar in that they both leverage the CLIP ViT-L/14 text encoder and a U-Net capable of processing $64 \times 64$ latent representations corresponding to $512 \times 512$ images. As Stable Diffusion v2.0 improves upon these with better text encoder and support for higher resolution images, we report the results for v2.0 on our Custom Concept and Landmarks setup in tables 5 and 6, respectively.

Table 5: Custom Concept results for Stable Diffusion v2.0 (avg. over 3 seeds)

| Method | $N_{param}$ Train($\downarrow$) | $N_{param}$ Store ($\downarrow$) | KID ($\downarrow$) (x $10^5$) | $A_{MMD}$ ($\downarrow$) (x $10^3$) | BwT$_{MMD}$ ($\uparrow$) | CLIP I2I ($\uparrow$) (x100) | $F_{MMD}$ ($\downarrow$) |
|---|---|---|---|---|---|---|---|
| C-LoRA | 0.05 | 100.29 | 158.9 | 114.5 | -103.88 | 68.10 | 0.015 |
| EWC | 0.05 | 100.29 | 141.3 | 100.97 | -96.50 | 75.81 | 0.011 |
| Ours: EWC DC | 0.05 | 100.29 | 134.97 | 93.59 | -88.30 | 77.29 | 0.0009 |
| Ours: DSC | 0.05 | 100.29 | 177.92 | 155.06 | -102.55 | 75.01 | 0.003 |
| Ours: DSC EWC | 0.05 | 100.29 | 139.76 | 93.71 | -91.02 | 77.90 | 0.006 |
| Ours: DSC EWC DC | 0.05 | 100.29 | 126.43 | 88.54 | -82.15 | 78.22 | 0.001 |

Table 6: Landmarks results for Stable Diffusion v2.0 (avg. over 3 seeds)

| Method | $N_{param}$ Train($\downarrow$) | $N_{param}$ Store ($\downarrow$) | KID ($\downarrow$) (x $10^5$) | $A_{MMD}$ ($\downarrow$) (x $10^3$) | $BwT_{MMD}$ ($\uparrow$) | CLIP I2I ($\uparrow$) (x100) | $F_{MMD}$ ($\downarrow$) |
|---|---|---|---|---|---|---|---|
| C-LoRA | 0.05 | 100.32 | 101.33 | 62.40 | -12.99 | 80.50 | 0.009 |
| EWC | 0.05 | 100.32 | 59.21 | 38.66 | -39.05 | 85.37 | 0.002 |
| Ours: EWC DC | 0.05 | 100.32 | 52.80 | 32.65 | -5.19 | 87.22 | 0.001 |
| Ours: DSC | 0.05 | 100.32 | 116.83 | 67.32 | -65.70 | 80.01 | 0.009 |
| Ours: DSC EWC | 0.05 | 100.32 | 59.40 | 45.27 | -5.71 | 86.88 | 0.0007 |
| Ours: DSC EWC DC | 0.05 | 100.32 | 46.15 | 29.13 | -2.90 | 88.14 | 0.084 |

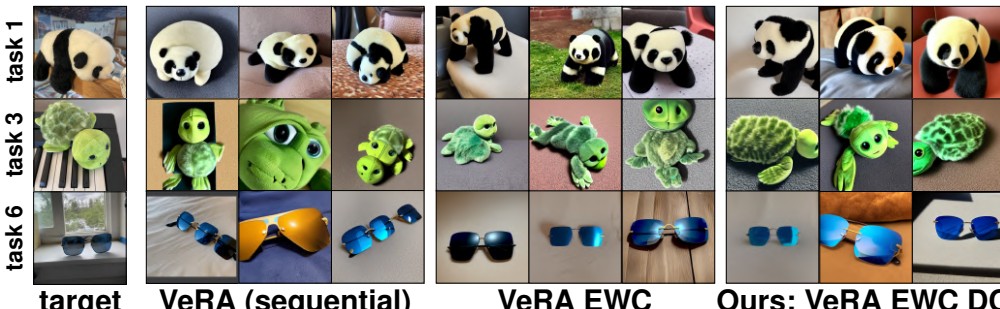

(a) **VeRA results:** we preserve our EWC DC framework and replace LoRA (Hu et al., 2022) with Vector-based Random Matrix Adapters (VeRA) (Kopiczko et al., 2024).

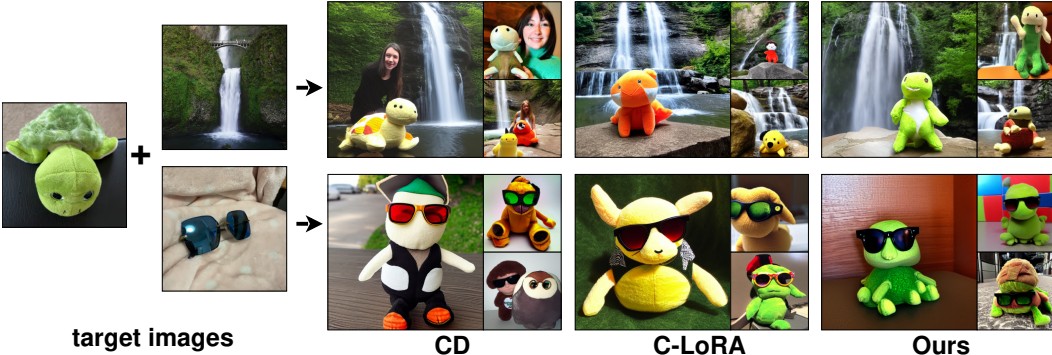

(b) **Multi-concept generation results:** the upper row images are generated using the prompt "A photo of $V^1$ plushie tortoise. Posing in front of $V^2$ waterfall" while the lower row images are generated using the prompt "A photo of $V^1$ plushie tortoise. Posing with $V^2$ sunglasses".

Figure 16: **Additional results** showing the compatibility of our method for (a) multi-concept generation, and (b) VeRA (Kopiczko et al., 2024).

## G.2 COMPATIBILITY WITH VERA

In the spirit of *parameter-efficient* continual personalization, we explore the effectiveness of our method for Vector-based Random Matrix Adaptation (VeRA) (Kopiczko et al., 2024). VeRA freezes the LoRA weight matrices $A$ and $B$ to share them across all network layers, and instead adapts two scaling vectors $\Lambda_b$ and $\Lambda_d$ per layer. This helps VeRA retain the performance of LoRA-based finetuning with a small fraction. For the U-Net, this amounts to a $\approx \frac{1}{100}$th reduction in the number of trainable parameters ($N_{param}$ Train) per task. We rely on the Diffusers library implementation of VeRA and use the default rank setup of 256. Fig. 16a compares the results of sequential VeRA, VeRA with EWC, and VeRA EWC with DC scores on our six task sequence of Custom Concept (Kumari et al., 2023). Here, VeRA sequential suffers from a loss of plasticity, e.g., sunglasses with three glasses, besides forgetting the precise details of previous tasks, e.g., distorted tortoise face/eyes in task 3. VeRA EWC helps improve over this despite struggling to retain knowledge at times, e.g., task 1 panda legs with white strips. Plugging in DC scores shows a clear gain in terms of the generative quality. The quantitative results are reported in App. table 7, and tell us a similar story.

Table 7: Custom Concept results with VeRA (Kopiczko et al., 2024) after 6 tasks (avg. over 3 seeds).

| Method | $N_{param}$ Train($\downarrow$) | $N_{param}$ Store ($\downarrow$) | KID ($\downarrow$) (x $10^5$) | $A_{MMD}$ ($\downarrow$) (x $10^3$) | BwT$_{MMD}$ ($\uparrow$) | CLIP I2I ($\uparrow$) (x100) | $F_{MMD}$ ($\downarrow$) |
|---|---|---|---|---|---|---|---|
| VeRA sequential | 0.0086 | 100.052 | 138 | 95.7 | -95.71 | **74.41** | 0.006 |
| VeRA EWC | 0.0086 | 100.052 | 145.2 | 98.27 | -98.73 | 73.37 | 0.0003 |
| Ours: VeRA EWC DC | 0.0086 | 100.052 | **140** | **94.7** | **-95.44** | 73.56 | **0.0001** |

### G.3 SUPPORT FOR MULTI-CONCEPT GENERATION

We study the compatibility of our proposed method for generating multiple custom concepts in the same picture. Figure 16b compares the multi-concept results of our method (LoRA EWC DSC DC) with that of C-LoRA (Smith et al., 2024b) and CD (Kumari et al., 2023). Following C-LoRA, we use the prompt style "a photo of $V^1$ [X]. Posing with $V^2$ [Y]", where $V^1$ and $V^2$ are the learnable custom tokens for the concept names X and Y, respectively. We find that for both the methods, multi-concept generation remains highly sensitive to prompt engineering, and minor prompt changes (replacing 'posing' with 'together') can lead to results where one of the concepts is overshadowed by another. We also notice similar effects upon removing the concept names X and Y from the prompt. On the other hand, using the concept names in the prompt can lead to interference from the model's pretrained knowledge, e.g., the waterfall in the background does not always resemble the waterfall from the target images. While both the methods struggle to produce images that preserve the exact style of the target images, we find that our method undergoes relatively lower interference than C-LoRA, which for instance, produces images of the plushy tortoise and sunglasses that differ significantly in color and appearance from the original target images.

Following our standard evaluation setup (Sec. 4), to quantify the performance for multi-concept generation, we generate 400 images each for the prompts used in Fig. 16b: "A photo of $V^1$ plushie tortoise. Posing in front of $V^2$ waterfall" and "A photo of $V^1$ plushie tortoise. Posing with $V^2$ sunglasses". In the absence of ground-truth images that incorporate multiple concepts, we quantify the scores by first relying on the ground truth images of the individual concepts in each of these prompts and then averaging out the two scores. Table 8 reports the scores averaged over the two multi-concept generation prompts. We note that these scores are in line with Fig. 16b where our EWC DSC DC variant performs significantly better than C-LoRA on all three performance quantification metrics.

Table 8: Results for multi-concept generation on Custom Concept setup

| Method | KID ($\downarrow$) (x $10^5$) | $A_{MMD}$($\downarrow$) (x $10^3$) | CLIP I2I ($\uparrow$) (x100) |
|---|---|---|---|
| CD | 231.7 | 194.11 | 48.50 |
| C-LoRA | 193.61 | 130.80 | 52.45 |
| Ours | **163.85** | **113.42** | **64.53** |

## H IMPLEMENTATION AND HYPERPARAMETERS

### H.1 IMPLEMENTATION DETAILS

We use the Hugging Face Accelerate library (Gugger et al., 2022) for distributed training/inference of our models. Our experiments are conducted using four RTX A6000 GPUs with 48 GB memory each. For all the compared methods, we set the batch size to 1 during training and inference. To allow for larger effective batch sizes during training, we set the gradient accumulation steps to 8. For a fair comparison with CD, we perform regularization during training using an auxiliary dataset of 200 images generated by the pretrained backbone using the prompt "a photo of a person" (Smith et al., 2024b). We follow C-LoRA for preserving a number of hyperparameters setup. Namely, we set the LoRA rank to 16, and the EWC loss coefficient to $1e6$ for all our experiments. We use a learning rate of $5e-6$ for all non-LoRA methods, and a learning rate of $5e-4$ for all LoRA-based methods. We implemented C-LoRA from scratch and following the authors, used a coefficient of

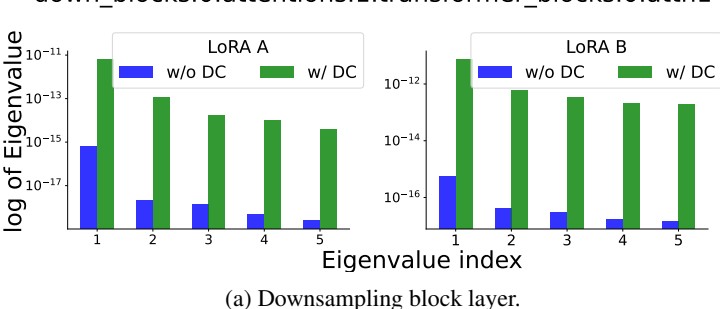

(a) Downsampling block layer.

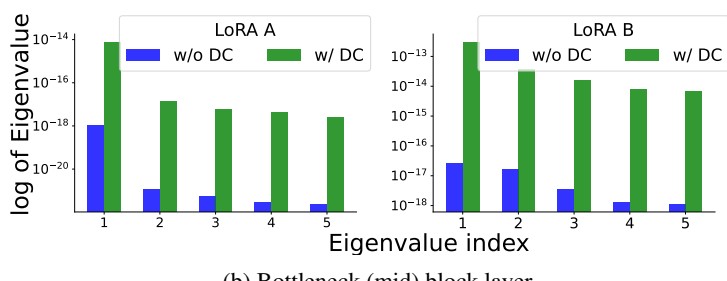

(b) Bottleneck (mid) block layer.

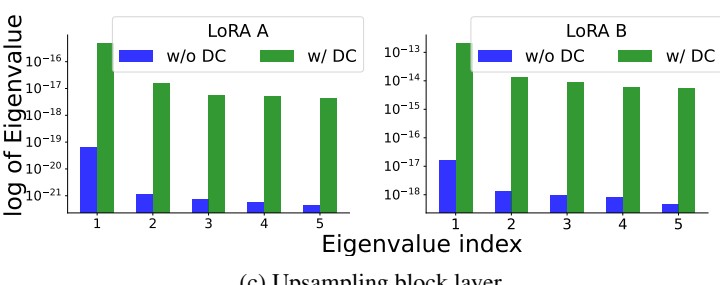

(c) Upsampling block layer.

Figure 17: **Sanity Check I for DC scores:** Top-k eigenvalue comparison for FIM estimated with and without DC scores using LoRA parameters for randomly chosen U-Net layers belonging to: (a) downsampling block, (b) mid block, and (c) upsampling block.

$1e8$ for the self-regularization loss $\mathcal{L}_{\text{forget}}$ (see Eq. 2). Lastly, following the default setup of the Diffusers library (von Platen et al., 2022), we set the classifier guidance scale to 7.5 to allow for a higher adherence to the conditional signal.

## H.2 IMPACT OF HYPERPARAMETERS

### H.2.1 NUMBER OF CONSOLIDATION ITERATIONS

We tune the number of consolidation iterations for EWC phase using our EWC DC variant and that for DSC phase using our EWC DSC DC variant. The number of consolidation iterations are fractions of the total training iterations, *i.e.,* 1000 on Custom Concept, and are chosen through a sweep on the set: $\{0.1\times, 0.2\times, 0.3\times, 0.5\times, 1\times\}$. As shown in Fig. 19 and 20, a value of $0.2\times$ the training iterations performs the best for both EWC DC and EWC DSC DC. While a larger number of iterations can lead to degradation in the generative quality of both the variants, we note that DSC remains more sensitive overall to the number of consolidation iterations.

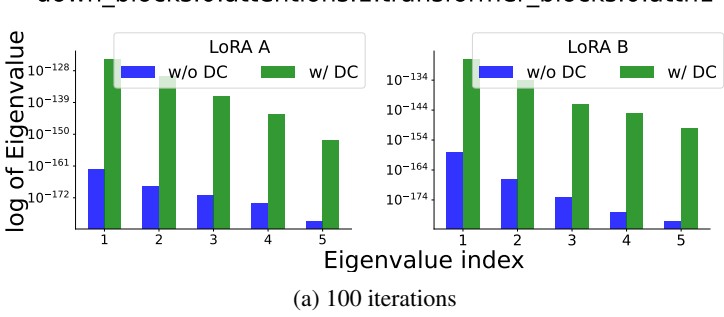

(a) 100 iterations

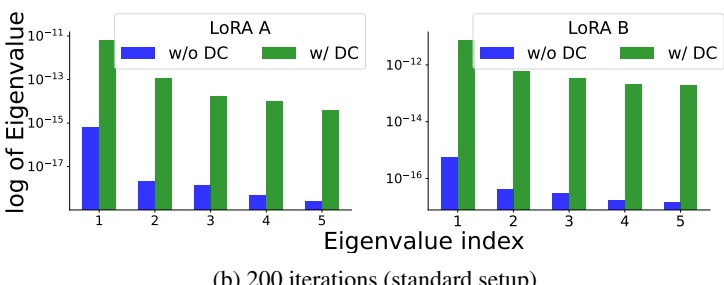

(b) 200 iterations (standard setup)

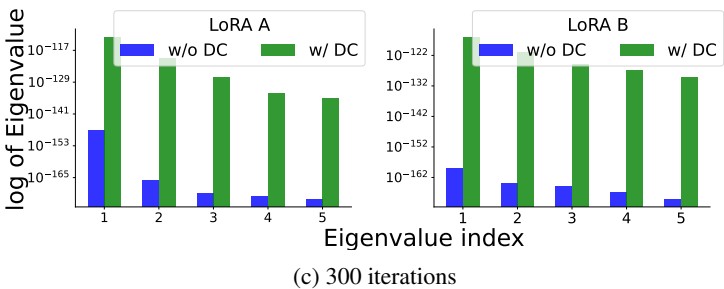

(c) 300 iterations

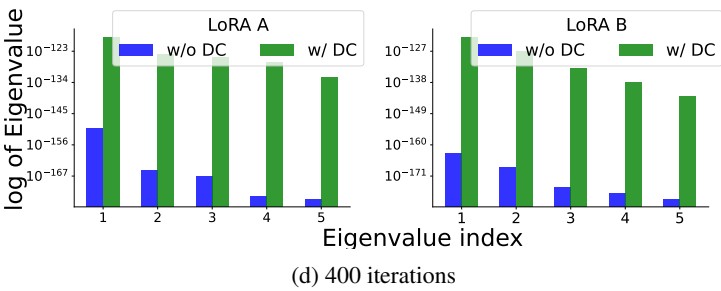

(d) 400 iterations

Figure 18: **Effect of varying consolidation iterations on FIM estimation:** Top-5 Eigenvalue comparison for the FIM estimated with and without DC scores using the learned LoRA parameters for a randomly chosen downsampling layer block of the U-Net. Upon increasing the number of consolidation iterations to 200, we observe larger magnitudes of Eigenvalues, thus indicating that the learned parameters become more certain about the directions of high loss changes from data, *i.e.*, reduced uncertainty in the FIM estimation. However, increasing the number of consolidation iterations beyond 200 leads to a saturation in the data-informed uncertainty estimation for the FIM.

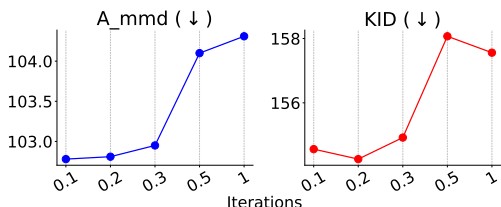

Figure 19: Effect of number of EWC iterations for EWC DC

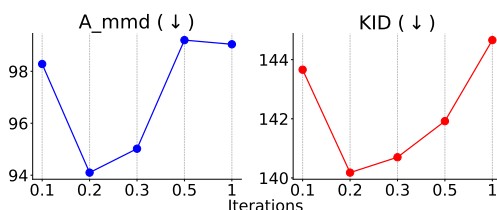

Figure 20: Effect of number of DSC iterations for EWC DSC DC

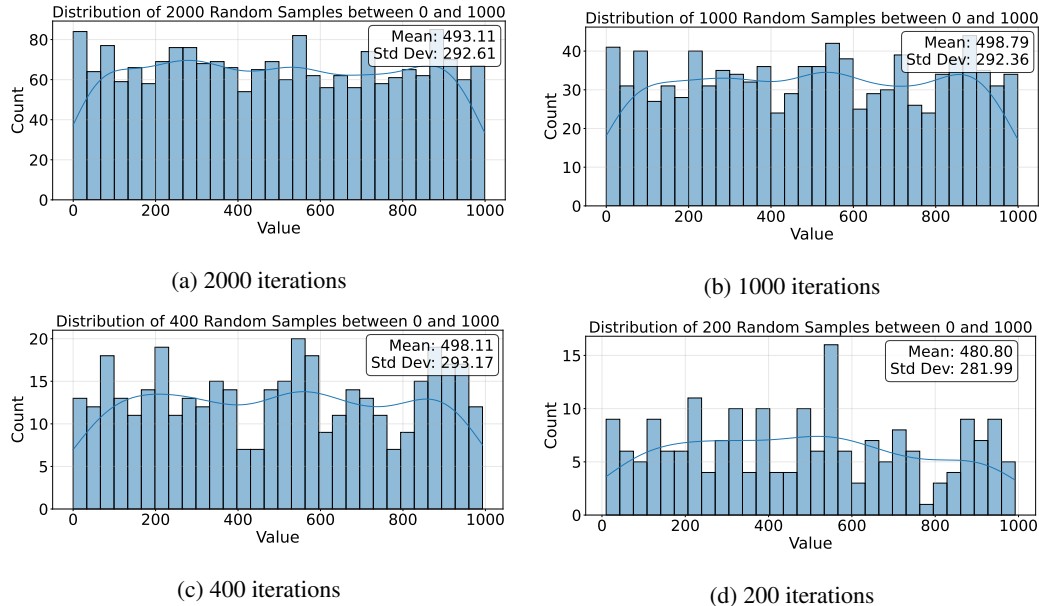

Figure 21: Distribution of uniformly sampled timesteps over a varying total number of training iterations: (a) 2000 finetuning iterations as used by Smith et al. (2024b) for Celeb-A faces, (b) 1000 finetuning iterations as used by Custom Diffusion Kumari et al. (2023) that we adopt for all but the Celeb-A setup, (c) 400 consolidation iterations that we adopt for the Celeb-A setup, (d) 200 consolidation iterations that we adopt for all but the Celeb-A setup. Note that the values for (c) and (d) are chosen per the hyperparameter tuning we perform in App. H.2.1.

### H.2.2    HYPERPARAMETER FOR EWC LOSS

Fig. 22 shows the impact of varying the hyperparameter $\delta$ controlling the contribution of the cross-entropy term for EWC DC (Eq. 5). We find the range $[0.5, 1.0]$ to be suitable for the loss weightage, and use $\delta = 0.5$ through our experiments.

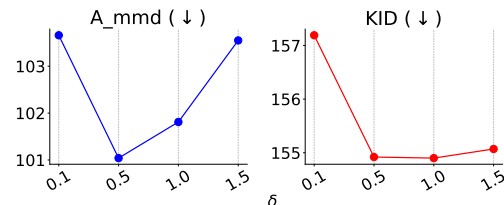

Figure 22: Effect of $\delta$ on performance of EWC DC

### H.2.3 HYPERPARAMETERS FOR DSC LOSS

**Effect of varying $\gamma$:** For DSC, $\gamma$ depicts the strength with which the student model $\theta_s$ follows the DC scores distribution of the $n^{\text{th}}$ task teacher $\theta_n$ and the previous task teacher $\theta_j$. As shown in Fig. 23, the range $[0.01, 0.1]$ remains suitable for $\gamma$. Accordingly, we set $\gamma$ to 0.1.

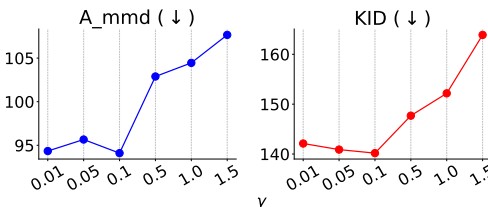

Figure 23: Effect of $\gamma$ on performance of DSC EWC DC

**Effect of varying $\lambda$:** $\lambda$ in DSC guides the strength with which the student $\theta_s$ matches the noise estimations of the $n^{\text{th}}$ task teacher $\theta_n$ and the previous task teacher $\theta_j$. Setting $\lambda = 0$ leaves the student consolidation to be guided solely by the discriminative DC scores, a setting that we find to be detrimental for the purpose of generation (see Fig. 10). In Fig. 24, we delve further into the impact of varying $\lambda$ on the performance of DSC EWC DC, and find the range $[0.5, 1.0]$ to work well. Note that a high $\lambda$ can lead the student to overfitting the noise estimations for the previous task teacher, for which the current task inputs remain out-of-domain. This, in turn, harms the generative quality of the student.

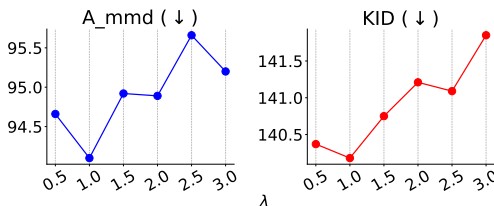

Figure 24: Effect of $\lambda$ on performance of DSC EWC DC

**Effect of varying the teacher's softmax temperature $\tau$:** Knowledge distillation frameworks typically rely on sharp target distributions that intuitively mimic the outputs of a confident teacher model (Caron et al., 2021). For our DSC framework, the teachers can be made to produce sharper targets by using a low value for the temperature $\tau$ in the softmax normalization operation of their DC scores. We show the impact of varying the teacher softmax temperature in Fig. 25. Namely, a temperature beyond 0.1 leads to softer targets from both the teachers, which can be harder to mimic for the student. On the contrary, $\tau = 0$ mimics extreme sharpening and produces one-hot hard distributions. We find a temperature of 0.05 to perform the best overall.

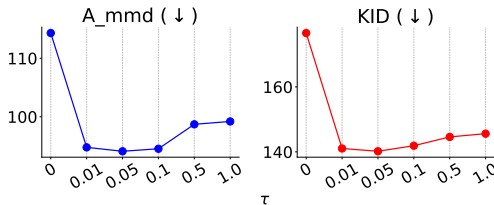

Figure 25: Effect of teacher softmax temperature $\tau$ on performance of DSC EWC DC

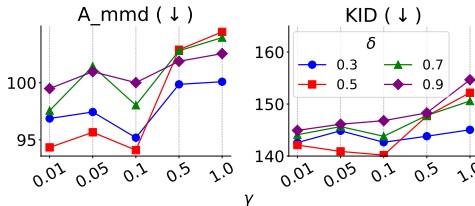

Figure 26: Effect of $\delta$ and $\gamma$ on performance of DSC EWC DC

### H.2.4  HYPERPARAMETER FOR EWC+DSC LOSS INTERACTIONS

## I  TRAINING TIME COMPLEXITY DERIVATION AND ANALYSES

We use the soft-O notation $\tilde{\mathcal{O}}$ (Van Rooij et al., 2019) to describe the time complexity while safely ignoring the logarithmic factors. Formally, for some constant $k$, $\tilde{\mathcal{O}}(f(n)) = \mathcal{O}(f(n) * \log^{k(n)})$ provides the upper bound for $f$, like the standard big-O notation $\mathcal{O}$ but hides the factors involving powers of logarithms, *i.e.*, $\tilde{\mathcal{O}}(n)$ could represent $\mathcal{O}(n \log n), \mathcal{O}(n \log \log n), \mathcal{O}(n \log^2 n)$, etc.

As also stated in the main paper, we consider a continual personalization setup with $N$ number of tasks such that each task comprises on new concept to acquire. For the ease of computation, we assume that each task has a fixed number of training images, $|\mathcal{D}|$. Note that for our CL setup, we use the same number of training epochs for each task. This lets us ignore the factor of training epochs in deriving the training time complexity. Lastly, since both C-LoRA (Smith et al., 2024b) and our setup train a single LoRA and a modifier token per task, their complexity of a forward pass remains the same, and can be safely ignored. Put together, we can state time complexity as a function that grows linearly with more training samples $|\mathcal{D}|$. We list the training time complexities of the compared methods in Table 9 and detail on their derivation below:

Table 9: Training time complexity analyses with soft-O notation $\tilde{\mathcal{O}}$.

| Method | Cost $\tilde{\mathcal{O}}(\cdot)$ |
| --- | --- |
| C-LoRA | $N \cdot |\mathcal{D}|$ |
| Ours: EWC DC | $|\mathcal{D}|$ |
| Ours: EWC DSC DC | $|\mathcal{D}|$ |

1. **C-LoRA** (Smith et al., 2024b) performs self-regularization using the weights of all previous task LoRA (see Eq. 2). Therefore, in addition to the training sample size, the time complexity of C-LoRA is dependent on the number of tasks $N$, *i.e.,* $\tilde{\mathcal{O}}(N|\mathcal{D}|)$.

2. **For parameter-space consolidation**, we rely on online EWC (Schwarz et al., 2018) which maintains a single set of FIM weights that are updated continuously using a running average over tasks. This ensures that our EWC-based framework does not store separate importance weights for each task, and hence, the time complexity scales linearly in the factor of sample size, $\tilde{\mathcal{O}}(|\mathcal{D}|)$. Next, for DC scores computation, we assume a fixed number of conditional forward passes that is proportional to the size of the relevant concept set $\mathbf{c}_k$ (see Sec. 3.1). Irrespective of the number of tasks, $\mathbf{c}_k$ always stores $m + 2$ number of concepts, where $m$ is chosen through grid search and is typically a low number for avoiding confusion

from other uninformative classes. Hence, DC scores computation for EWC scales linearly with the number of training samples $\tilde{\mathcal{O}}(|\mathcal{D}|)$. Put together, the time complexity for our parameter-space consolidation framework is: $\tilde{\mathcal{O}}(|\mathcal{D}|) + \tilde{\mathcal{O}}(|\mathcal{D}|) = \tilde{\mathcal{O}}(|\mathcal{D}|)$.

3. **For function-space consolidation**, we rely on a double-distillation framework, which uses two teacher and one student LoRA per consolidation iteration, irrespective of the number of seen tasks. Subsequently, the training time complexity of function-space consolidation remains $\tilde{\mathcal{O}}(|\mathcal{D}|)$. For computing DC scores, we always rely on three conditional forward passes through each of the teachers and the student. As described in Sec. 3.2, these forward passes correspond to the readily available common prior concept $c_0$, and the concepts $c_n$ and $c_{j<n}$ corresponding to the current task $n$ and the previous task $j < n$ teacher LoRA. Hence, the time complexity of DC scores computation is also $\tilde{\mathcal{O}}(|\mathcal{D}|)$. Overall, the time complexity for our function-space consolidation framework remains: $\tilde{\mathcal{O}}(|\mathcal{D}|) + \tilde{\mathcal{O}}(|\mathcal{D}|) = \tilde{\mathcal{O}}(|\mathcal{D}|)$.

**Runtime per training iteration.** While our method scales better than C-LoRA (Smith et al., 2024b) with the number of tasks (see Table 9), we are nevertheless bounded by the several conditional forward passes needed (depending on the value of $k$) to derive DC scores for each training minibatch. Despite our proposed considerations for efficient computation of DC scores during consolidation (see Sec. 3.1), the computational overhead for deriving these remains dominant specifically for CL setups with fewer number of tasks. For example, using an RTX A6000, each consolidation iteration for EWC requires $\approx 5.3$s, that for DSC requires $\approx 5.7$s, and that for C-LoRA requires $\approx 0.8$s during finetuning on the task 6 of our Custom Concept setup. As shown in Table 10, scaling to the 50 tasks setup, this time gap bridges as the runtime per training iteration of C-LoRA grows to $\approx 3.8$s while that of our methods stay roughly the same ($\approx 5.5$s for EWC, and $\approx 5.72$s for DSC for $k = 5$). It is worth noting that on the $50^{\text{th}}$ training task, for lower values of $k$, the runtime per training iteration for our methods remain comparable (for $k = 3$) or significantly lower (for $k = 2$) than that of C-LoRA. Therefore, we expect that more clever ways to derive the DC scores during training can effectively reduce the runtime of our methods.

Table 10: Comparison of training time per iteration (wall clock time in seconds) with varying $k$

| Method | k = 2 | k = 3 | k = 5 |
|---|---|---|---|
| On $6^{\text{th}}$ task | | | |
| C-LoRA | | 0.8 | |
| Ours: EWC DC | 1.27 | 3.4 | 5.3 |
| Ours: DSC DC | 1.65 | 3.61 | 5.7 |
| On $50^{\text{th}}$ task | | | |
| C-LoRA | | 3.8 | |
| Ours: EWC DC | 1.36 | 3.57 | 5.5 |
| Ours: DSC DC | 1.8 | 3.66 | 5.72 |

## J FAILURE CASES

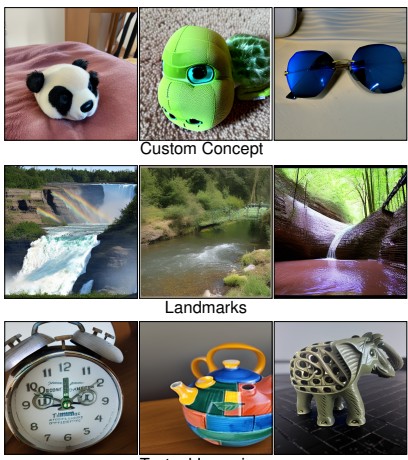

Figure 27: **Failure cases** showing the visual artefacts of our best performing variant EWC DSC DC on our three different dataset setups.

Despite our method retaining significantly better task-specific generation granularity compared to the state-of-the-art, it produces noticeable visual artefacts sometimes. Fig. 27 shows few such dataset-specific artefacts for EWC DSC DC, which is our overall best performing variant leveraging DC scores with EWC and DSC. Notably, for Custom Concept, the model at times generates figures that have out-of-proportion shapes including an absence of the plushie panda's body (left), an unnaturally big head for the plushie tortoise (middle), and a poorly outlined frame for the wearable sunglasses (right). For the waterfall landmarks setup, we notice multiple incomplete rainbows (left), a transparent yet poorly formed bridge over the river (middle), and a mulberry colored waterfall foreground (right). Similarly, on the textual inversion setup, the generated clock image has incorrectly printed numers (left, with 11 replacing 1 and 3 being confused with 9), the teapot with incorrectly assigned spout/handles (middle), and the elephant's body with holes that have unnaturally filled background.

