# OpenReview forum: "Mining your own secrets: Diffusion Classifier Scores for Continual Personalization of Text-to-Image Diffusion Models"
_ICLR.cc/2025/Conference — ICLR 2025 Poster_

### Official Review · Reviewer_w18g · 2024-11-04

**Soundness:** 3
**Presentation:** 3
**Contribution:** 3
**Rating:** 6
**Confidence:** 4

**Summary:**

This paper focuses on the continual personalization of pretrained text-to-image diffusion models. It aims to address the challenge of balancing new concept learning and previous concept forgetting
using a two level optimization strategy, i.e Deep Model Consolidation (DMC) and Elastic Weight Consoliation (EWC) . Buliding on DMC and EWC, the author introduce Diffusion Classifier (DC) as an additional constrait during consolidation.

**Strengths:**

1. The key innovation of this paper is the combination of DC scores with the EWC algorithm. This provides an interesting approach to applying EWC in T2I diffusion models.
2. The paper is well-written, with a clear and detailed explanation of the preliminary concepts in the related work section. The discussion on the limitations of C-LoRA naturally leads to introducing the EWC algorithm and the use of DC for likelihood estimation.

**Weaknesses:**

1. The optimization for the $n_{th}$ task combines a function-space objective with DSC (Eq. 6) and a parameter-space objective using EWC, but the paper lacks a detailed objective function that includes the FIM. This makes it difficult to understand how DSC and EWC work together during optimization. Consider adding extra consolidating Algorithms 1, 2, and 3 into a summary algorithm that corresponds to Fig. 2 to improve clarity.
2. In Fig. 4 and Table 2, results show that EWC + DC and DSC + EWC + DC achieve comparable performance. Given that the DSC algorithm introduces complexity similar to that of EWC + DC, this raises questions about the necessity of DSC’s design.
3. A key limitation of this method is computational complexity. Each iteration of DSC and EWC + DC requires multiple complete diffusion forward and backward passes (Eq. 4 and Eq. 6). In contrast, C-LoRA’s explicit regularization is more efficient. It is recommended to further discuss the complexity of a single iteration and to provide a quantitative description of the complexity reductions achieved through pruning optimizations, such as selecting a subset of seen concepts (Fig. 16).

**Questions:**

In Algorithm 1, lines 5-6, the update of the DC scores dictionary is unclear. Line 6 shows that $P_\theta[c^i]$ is normalized for a subset with a coordinate of $k$, while line 5 indicates it is a probability distribution over $n + 1$ concepts.

---

> ### Author Response · Authors · 2024-11-18
>
> Dear Reviewer w18g,
>
> Thank you for your comments and suggestions. We have tried our best to address your concerns below.
>
> 1. **W.1: A summary algorithm to improve clarity:** Thank you for your suggestion. We have accordingly updated the pdf with a summary Algorithm 4 in Appendix A that shows how DSC and EWC work together during optimization using Algorithms 1, 2, and 3 as references.
>
> 2. **W.2: On the necessity of DSC:** Thank you for your comment. **First,** from an empirical viewpoint, we note that DSC + EWC + DC outperforms EWC + DC on all our setups across majority of the metrics. The performance gain of the former over the latter varies across setups with DSC + EWC + DC surpassing EWC + DC significantly on the setups of Custom Concept (Table 1), Textual Inversion (Table 3), and Celeb-A (Table 4). Our additional experiments on Stable Diffusion v2.0 (mentioned in the rebuttal to Reviewer 76RN) also show consistent performance gains of DSC + EWC + DC over EWC + DC. **Second,** from an intuitive viewpoint, EWC is not enough if we want to retain knowledge from some specific previous task LoRA . This is because with EWC, we maintain a task-shared Fisher information matrix that continuously updates the importance scores of LoRA parameters based on the training data from each new task. On the other hand, with the DSC framework, we can be more flexible at controlling which task’s knowledge do we want to distill more. This can simply be achieved by replacing the uniform sampling for teacher-2 with a weighted sampling strategy such that the desired task’s LoRA has a higher chance of being sampled as teacher-2 (on line 5, Algorithm 3 in the main paper). As such, DSC can be clearly seen to be adding performance and flexibility gains to our consolidation framework.
>
> 3. **W3: On the computational complexity and comparison with C-LoRA:** **First**, we argue that C-LoRA is problematic in Sec 3.1, and hence is not a suitable choice for parameter-space consolidation – our results across all dataset setups show that C-LoRA exhibits catastrophic forgetting across several tasks. **Second**, per your suggestion, in Table 10 (Appendix I) in the updated pdf, we have compared the wall clock time for a single iteration of C-LoRA with our proposed algorithms using different values of the subset of seen concepts **k** on the latter. In line with our theoretical training time complexity analyses from Sec. 4.3, we find that as the number of tasks grow, the training time gap per iteration between C-LoRA and our proposed methods bridges. For instance, on the 50th training task, the time required per training iteration of our proposed algorithms is almost half of that of C-LoRA's, e.g., for k = 2, C-LoRA requires 3.8s while EWC + DC requires 1.36s and DSC + DC requires 1.8s. For the ease of readability, we provide the Table 10 comparison results of training time per iteration (wall clock time in seconds) below too:
>
> | **Method**          | k = 2 | k = 3 | k = 5 |
> |----------------------|-------|-------|-------|
> | **On $6^{\text{th}}$ task** |       |       |       |
> | C-LoRA              | 0.8 | 0.8 | 0.8 |
> | Ours: EWC DC        | 1.27  | 3.4   | 5.3   |
> | Ours: DSC DC        | 1.65  | 3.61  | 5.7   |
> | **On $50^{\text{th}}$ task** |       |       |       |
> | C-LoRA              | 3.8 | 3.8 | 3.8 |
> | Ours: EWC DC        | 1.36  | 3.57  | 5.5   |
> | Ours: DSC DC        | 1.8   | 3.66  | 5.72  |
>
> 4. **Q.1: On the updates of $p_\theta[c^i]$ in Algorithm 1:** Thank for your comment. **$p_\theta$** is indeed a probability distribution over *n+1* concepts (*n* personalization concepts and one pre-trained concept). However, as mentioned in Line 5 Algo. 1, for one consolidation iteration, we only seek to update the probability values for *k* out of *n+1* concepts, where *k* is the subset of seen concepts that are relevant for computing DC scores (shown in lines 4 and 9 in Algo. 2 and Algo. 3, respectively). As a result, even though Line 5 in Algo 1 initializes probability values for all *n+1* concepts with a very small non-zero score of $ω$ = 1e-10, it is only the *k* concepts for which the probability scores are later updated (to a larger value) on line 6.
>
> Hence, initialization of DC scores dictionary serves two-fold purposes. **First**, initializing the probability scores for all *n+1* concepts ensures that the model gets to see a consistently ordered probability distribution for all *n+1* concepts across iterations. **Second**, using a very small yet non-zero value for $ω$ helps us avoid any potential numerical instability that could result from taking the logarithm of the zero-valued probability scores while computing the cross-entropy losses for EWC (Algo. 2 , line 10) and DSC (Algo 3, line 12).

---

### Official Review · Reviewer_76RN · 2024-11-04

**Soundness:** 3
**Presentation:** 3
**Contribution:** 2
**Rating:** 6
**Confidence:** 4

**Summary:**

This paper introduces a novel approach for continual personalization of text-to-image diffusion models using Diffusion Classifier (DC) scores. The method focuses on integrating DC scores as regularizers to mitigate the problem of forgetting in continual learning setups. By employing parameter-space and function-space regularization techniques, the authors aim to maintain previously acquired knowledge while integrating new concepts. The approach is evaluated against several baselines across multiple datasets and scenarios, demonstrating improved performance in retaining learned concepts with minimal parameter overhead.

**Strengths:**

1. The motivation of the paper is straightforward and clear.
2. The paper is technically sound, presenting a well-structured approach for continual personalization of text-to-image diffusion models using Diffusion Classifier (DC) scores.
3. The method seems computationally efficient, supported by sufficient experimental results and details.

**Weaknesses:**

**Strengths:**

1. The motivation of the paper is straightforward and clear.
2. The paper is technically sound, presenting a well-structured approach for continual personalization of text-to-image diffusion models using Diffusion Classifier (DC) scores.
3. The method seems computationally efficient, supported by sufficient experimental results and details.

**Weaknesses:**

1. While the paper is technically proficient, the novelty of this paper seems limited. The primary innovation appears to be the use of diffusion scores as a regularization term for continual personalization, which, while interesting, is not a significant departure from existing methods. The core of the approach builds upon well-established techniques such as Elastic Weight Consolidation, adapting them for diffusion models rather than introducing fundamentally new methodologies.
2. Since Stable Diffusion versions 1.5, 2.0, 2.1, and XL have already been released, why not use the newer versions? At the very least, incorporating some of them would demonstrate that your method can be easily generalized across different architectures.
3. The paper would be more interesting if it could test on more recently proposed personalization methods like LyCORIS [1] and DoRA [2].
4. Since multiple-concept generation is common when verifying the effectiveness of proposed personalization methods according to your cited papers [3,4], could the authors further provide experimental results on multi-concept generation results?

[1] Yeh, Shih-Ying, Yu-Guan Hsieh, Zhidong Gao, Bernard BW Yang, Giyeong Oh, and Yanmin Gong. "Navigating text-to-image customization: From lycoris fine-tuning to model evaluation." In *The Twelfth International Conference on Learning Representations*. 2023.

[2] Liu, Shih-Yang, Chien-Yi Wang, Hongxu Yin, Pavlo Molchanov, Yu-Chiang Frank Wang, Kwang-Ting Cheng, and Min-Hung Chen. "Dora: Weight-decomposed low-rank adaptation." *arXiv preprint arXiv:2402.09353* (2024).

[3] Ruiz, Nataniel, Yuanzhen Li, Varun Jampani, Yael Pritch, Michael Rubinstein, and Kfir Aberman. "Dreambooth: Fine tuning text-to-image diffusion models for subject-driven generation." In *Proceedings of the IEEE/CVF conference on computer vision and pattern recognition*, pp. 22500-22510. 2023.

[4] Kumari, Nupur, Bingliang Zhang, Richard Zhang, Eli Shechtman, and Jun-Yan Zhu. "Multi-concept customization of text-to-image diffusion." In *Proceedings of the IEEE/CVF Conference on Computer Vision and Pattern Recognition*, pp. 1931-1941. 2023.

**Questions:**

see above

---

> ### Author Response · Authors · 2024-11-17
>
> Dear Reviewer 76RN,
>
> Thank you for your comments and suggestions. We have tried our best to address your concerns below.
>
> 1. **On the novelty of our work:** Thank you for your remark. While our approach does build upon techniques like Elastic Weight Consolidation, the novelty of our work lies in the innovative application of diffusion scores as a regularization term for continual personalization in diffusion models—a connection not previously explored. From a *broader standpoint*, our approach advocates that going beyond the previous applications for test-time classification [1,2], the pre-trained class-specific representation knowledge of diffusion models can further be reinforced through thoughtfully designed finetuning frameworks which in turn can help enhance the performance of downstream generation tasks. Given that our thorough empirical analyses validate the effectiveness of our method and that no other work has explored the utility of diffusion classifier scores for enhancing continual personalization of diffusion models, we believe that our approach is **ipso facto** novel [3].
>
> 2. **Experiments with Stable Diffusion 2.0:** As we follow the setup for C-LoRA paper, which in turn leverages the Custom Diffusion paper setup [4] for training, we choose Stable Diffusion v1.4 to reproduce their setting as closely as possible. Also, we note that v1.4 and v1.5 remain architecturally very similar in that they both leverage the CLIP ViT-L/14 text encoder and a U-Net capable of processing 64x64 latent representations corresponding to 512x512 images. As Stable Diffusion v2.0 improves upon these (with better text encoder and support for higher resolution images), we report the results for v2.0 on our Custom Concept setup here as well as in Appendix Table 7 in the updated pdf:
>
> | Method              |  KID ($\downarrow$) (x $10^5$) | $A_{\text{MMD}}$ ($\downarrow$) (x $10^3$) | $\text{BwT}_{\text{MMD}}$ ($\uparrow$) | CLIP I2I ($\uparrow$) (x100) | $F_{\text{MMD}}$ ($\downarrow$) |
> |-----------------------|------------------------------|--------------------------------------------|----------------------------------------|------------------------------|--------------------------------|
> | C-LoRA              | 158.9                        | 114.5                                      | -103.88                                | 68.10                        | 0.015                          |
> | EWC                 | 141.3                        | 100.97                                     | -96.50                                 | 75.81                        | 0.011                          |
> | Ours: EWC DC        | 134.97                       | 93.59                                      | -88.30                                 | 77.29                        | **0.0009**                         |
> | Ours: DSC            | 177.92                       | 155.06                                     | -102.55                                | 75.01                        | 0.003                          |
> | Ours: DSC EWC        | 139.76                       | 93.71                                      | -91.02                                 | 77.90                        | 0.006                          |
> | Ours: DSC EWC DC      | **126.43**                       | **88.54**                                      | **-82.15**                                 | **78.22**                        | 0.001                          |
>
>
> References:
>
> [1] Li *et al.* "Your Diffusion Model is Secretly a Zero-Shot Classifier”. ICCV 2023.
>
> [2] Clark *et al.* "Text-to-Image Diffusion Models are Zero-Shot Classifiers". NeurIPS 2023.
>
> [3] "Novelty in Science," Medium blog post by Michael Black, [https://medium.com/@black_51980/novelty-in-science-8f1fd1a0a143](https://medium.com/@black_51980/novelty-in-science-8f1fd1a0a143)
>
> [4] Custom Diffusion results are based on **Stable diffusion v1.4**: [https://github.com/adobe-research/custom-diffusion?tab=readme-ov-file#results](https://github.com/adobe-research/custom-diffusion?tab=readme-ov-file#results)

---

> ### Author Response · Authors · 2024-11-18
>
> 2. **Experiments with Stable Diffusion v2.0 (Continued):** We report the results for v2.0 on our Waterfall landmark setup here as well as in Appendix Table 8 in the updated pdf:
>
> | Method                |  KID ($\downarrow$) (x $10^5$) | $A_{\text{MMD}}$ ($\downarrow$) (x $10^3$) | $\text{BwT}_{\text{MMD}}$ ($\uparrow$) | CLIP I2I ($\uparrow$) (x100) | $F_{\text{MMD}}$ ($\downarrow$) |
> |-----------------------|------------------------------|--------------------------------------------|----------------------------------------|------------------------------|--------------------------------|
> | C-LoRA               | 101.33                       | 62.40                                      | -12.99                                 | 80.50                        | 0.009                          |
> | EWC                  | 59.21                        | 38.66                                      | -39.05                                 | 85.37                        | 0.002                          |
> | Ours: EWC DC         | 52.80                        | 32.65                                      | -5.19                                  | 87.22                        | 0.001                          |
> | Ours: DSC            | 116.83                       | 67.32                                      | -65.70                                 | 80.01                        | 0.009                          |
> | Ours: DSC EWC        | 59.40                        | 45.27                                      | -5.71                                  | 86.88                        | **0.0007**                         |
> | Ours: DSC EWC DC    | **46.15**                        | **29.13**                                      | **-2.90**                                  | **88.14**                        | 0.084                          |
>
> From the above results, we note that our variant using Diffusion Classifier (DC) scores with EWC and DSC performs consistently the best on both the dataset setups (highlighted in bold) with Stable Diffusion v2.0.
>
> 3. **Results on all Stable Diffusion versions and on more parameter-efficient finetuning methods:** We would like to state that text-to-image diffusion is a rapidly evolving field, as is the field of parameter-efficient finetuning itself. Given this fast-paced environment, it is challenging to provide an exhaustive comparison with every single version of the Stable diffusion model and every recent parameter-efficient finetuning method. While we strive for comprehensive analysis, we hope the reviewer understands that an all-encompassing comparison may not be practically achievable within the scope of this work. That said, we have reported the results with VeRA [1] at the end of Sec. 4.3 in the main paper, since it has a more flexible support within the Diffusers library than LyCORIS and DoRA.  To summarize, our analyses demonstrate the effectiveness of our proposed method for two Stable Diffusion versions: **v1.4** and **v2.0**, and for two parameter-efficient finetuning adapters: **LoRA** and **VeRA**.
>
> 4. **Further experimental results on multi-concept generation:** Thank you for your comment. Per your suggestion, we have updated the paper with the results for sequentially trained Custom Diffusion (CD) on multi-concept generation (see Fig. 17b in Appendix G.2). We have also provided the *quantitative scores* for multi-concept generation in Table 6, and have detailed (in blue font) on *how* we compute these scores (given that we do not have ground truth images that incorporate multiple concepts) in Appendix G.2. In line with our main paper findings (Sec. 4.1) on single-concept generation, we see that CD has forgotten the custom category attributes (color, background, etc.) and instead generates multi-concept images that exhibit the model’s pretrained knowledge about the subject (e.g. humans posing with the plushie tortoise in front of the waterfall). For the ease of readability, we mention the quantitative results of Table 6 below (bold refers to the best scores):
>
> | Method   | KID ($\downarrow$) (x $10^5$) | $A_{\text{MMD}}$ ($\downarrow$) (x $10^3$) | CLIP I2I ($\uparrow$) (x100) |
> |----------|--------------------------------|-------------------------------------------|-----------------------------|
> | CD       | 231.7                          | 194.11                                    | 48.50                      |
> | C-LoRA   | 193.61                         | 130.80                                    | 52.45                      |
> | Ours     | **163.85**                     | **113.42**                                | **64.53**                  |
>
> **References:**
>
> [1] Kopiczko *et al.*, "VeRA: Vector-based Random Matrix Adaptation". ICLR 2024.

---

> > ### Comment · Reviewer_76RN · 2024-11-21
> > **thanks for your reponses**
> >
> > Thanks for your responses. My concerns have been addressed. I vote for a weak acceptance.

---

### Official Review · Reviewer_CPEf · 2024-11-05

**Soundness:** 3
**Presentation:** 2
**Contribution:** 2
**Rating:** 6
**Confidence:** 3

**Summary:**

The paper presents a novel framework for continual personalization (CP) in text-to-image diffusion models using Elastic Weight Consolidation (EWC) and Diffusion Scores Consolidation (DSC). A key part of their approach is using DC scores for regularization, which leverages class-specific information from the diffusion model.

**Strengths:**

- The use of diffusion classifier for continual learning is novel and interesting.
- The limitation of the previous work is well addressed.
- Related work is well summarized.

**Weaknesses:**

- Variance in approximation of expectation: During consolidation and updating FIM, the expectation over DC scores is approximated with a single trial per minibatch. While the authors aim to mitigate this by averaging across epochs, this can induce a biased Monte Carlo estimate. More rigorous analysis or empirical evidence is needed to demonstrate that this approximation does not compromise stability or lead to noisy FIM updates.
- Loss Component Interactions: The paper introduces multiple loss components, which can make hyperparameter tuning challenging. While the authors conducted an ablation study to evaluate the impact of individual losses, they did not address how these losses interact with each other. A more in-depth analysis of the interactions between different loss terms would strengthen the paper and provide clearer insights into optimizing the overall training process.
- The link between classification and generative quality: For **Sanity Check II**, the paper lacks sufficient detail regarding the experimental setup. It does not clearly define what classifier is being used or how the classification is being performed (Is that diffusion classifier accuracy with a single trial?).
    - Additionally, I don’t think that improved classification alone does not necessarily imply that the quality of generated images will also improve. There needs to be a clear demonstration of the connection between accurate classification and enhanced image generation to support this claim.
- Selection of $k$ (number of sampled tasks) for figure 16**:** I think choosing the number of sampled tasks  $k$  is crucial for achieving good results. However, the paper does not provide a clear strategy for determining the optimal  $k$ . While it shows that selecting five tasks works best in an experimental setting with six total tasks, it does not offer guidance on how to select $k$ when the total number of tasks $n$ changes.
Editorials: The notation for class labels is somewhat unclear throughout the manuscript. c is introduced as a text prompt in line 103, used as a class label in line 146, and used as a one-hot label in line 245. Since this notation is widely used throughout the manuscript, it is important to clarify its definition in the first place. Using superscript as an index is confusing, especially with exponentiation.
- Minor comments
    - Equation 5 needs to be fixed. Since L_denoise is an expectation over data, noise, concept, and time steps, the second term in r.h.s. of Eq (5) needs to be an expectation over data.
    - The title of the manuscript is somewhat misleading. What does secret mean in this context?

**Questions:**

- In explaining the limitation of C-LoRA, it is said that “L_forget decreases throughout training, thus losing most of the information learned for task1”. If I’m not mistaken, no forgetting happens when L_forget is close to zero since the modified parts of the parameters do not overlap with the previous LoRA parameters. Is there something I’m missing?
- Is there a difference between the results in Section 3.1 and the proposed method? Specifically, could you provide the results in Figure 1 for the proposed method? I’m curious if the changes in weight and loss differ from those observed in C-LoRA.
- In line 172, the authors mention learning a new word vector  $V_n$ , but neither the algorithm nor the figure includes this  $V_n$ , which is important for personalization. It only appears to be used during evaluation. This creates confusion about how  $V_n$  is obtained and when it is used in the training process.
- Including specific examples of pre-trained concept $c_0$$c$  and  $V_n$  would make it easier to understand.

---

> ### Author Response · Authors · 2024-11-17
>
> Dear Reviewer CPEf,
>
> Thank you for your comments and suggestions. We have tried our best to address your concerns below.
>
> 1. **W.1: Variance in approximation of expectation:** While single trial per batch may introduce some bias in Monte Carlo estimate, this bias is averaged out over many batches, as each timestep has roughly an equal probability of being selected. Accordingly, we provide Fig. 22 in our updated pdf to show that our number of consolidation iterations (400 for the Celeb-A setup and 200 for the rest setups) remain enough to ensure that all timesteps are represented. One special case we note is that for fewer number of consolidation iterations (Fig. 22d), the intermediate timesteps have a relatively higher sampling chance. However, under limited timestep evaluations per class, previous works have shown that diffusion classification scores remain most sensitive to the conditioning signal for the intermediate timesteps [1]. Hence, a higher concentration of probability mass on the intermediate timesteps favors classification accuracy when the number of consolidation iterations remains limited.
>
>  - Lastly, our **empirical results** in App. H.2.1. show that the generative quality (quantified in terms of Average MMD and KID scores) first increases as the number of consolidation iterations is increased to roughly 0.2 times that of the total training iterations and then saturates.
>
>  - **More analyses for FIM updates:** Per your suggestion, in Fig. 19 in the updated pdf, we compare the effect of varying the number of consolidation iterations on FIM estimation. Namely, we observe that as the number of consolidation iterations increase from 100 to 200 leads to larger magnitudes of top-5 Eigenvalues, thus indicating that the learned parameters become more certain about the directions of high loss changes from data, i.e., reduced data-driven uncertainty in the FIM estimation. However, increasing the number of consolidation iterations beyond 200 does not lead to any further growth in the top-k Eigenvalue magnitudes thus indicating a saturation point in data-informed uncertainty estimation for the FIM. We note that this observation is in line with 200 being our optimal number of consolidation iterations on the Custom Concept setup found via hyperparameter sweep  (Appendix H.2.1).
>
> 2. **W.2: Loss component interactions:** We are sorry for the confusion caused by a lack of an explicit study on how the performance varies with the loss component interactions. We have updated the pdf with Appendix H.2.4 to show the effect of varying the hyperparameters δ and γ that control the strength of matching the DC scores in the loss terms for EWC (Eq. 5) and DSC (Eq. 6), respectively. In particular, we find that the performance variation of the DSC EWC DC model due to changes in [$δ$, $γ$] hyperparameters is consistent with the individual performance changes due to variations in $δ$  and $γ$ as reported in Fig. 23 and 24, respectively. That is, we find that a value close to 0.1 remains optimal for $γ$ while $δ = 0.5$ performs the best. Moreover, a larger $γ$ remains detrimental for the purpose of generation, marked by consistently higher KID and A_MMD scores. This observation is consistent with that seen in Fig. 24 given that a larger $γ$ can lead to the student consolidation to be influenced more by the discriminative DC scores and lesser by the denoising score matching score. This in turn harms the generation quality of the images, with an extreme case shown in Fig. 11 where the generated images can turn unintelligible in case of zero guidance of denoising scores during consolidation.
>
> 3. **W.3: The link between classification and generative quality:** For Sanity Check II, we classify the training set points using the diffusion classifier (DC) scores with a single trial and the 500th timestep following [1]. You are correct that improved classification alone does not imply an improvement in the generation quality. **Instead**, as mentioned in Sec. 4.3, this sanity check is designed to ensure that our consolidated model **does** leverage DC scores. Namely, if the improvement in the generation quality of the model comes from consolidation losses other than the DC scores, i.e., either the DC scores are not being leveraged during consolidation or the derived DC scores are unreliable, then we do not expect a significant improvement in the training set classification (as can be seen with EWC and C-LoRA). On the other hand, if the model trained on all tasks has been consolidated with reliable DC scores from their training datasets, then we expect it to improve on the classification scores of these training samples.
>
> **References:**
>
> [1] Li *et al.*, "Your Diffusion Model is Secretly a Zero-Shot Classifier". ICCV 2023.

---

> ### Author Response · Authors · 2024-11-17
>
> 4. **W.4: Selection of *k* (number of sampled tasks) for Figure 16:** We first note that our consolidation stages for *n*-th task leverage only the *n*-th task training data, *i.e.*, in a replay-free setup. As such, we must use the *n*-th task LoRA and its corresponding *n*-th task concept **$c_n$**, i.e. *k* = 1. Next, following Custom Diffusion and C-LoRA, we also leverage the readily available pretrained concept **$c_0$** that each LoRA has learned, *i.e.*, *k* = 2. For *k* = 3 onward, we find that increasing *k* up to a certain point helps improve the generation quality as the DC scores carry more discriminative information with more classes. Namely, we find that  the generative quality saturates with *k* > 5 since not all previous concepts carry useful discriminative information for classification. Hence, an intermediate value of *k* helps strike a good balance between discriminative information and inter-class confusion. Lastly, although we perform this selection of *k* on the six task setup of the Custom Concept dataset, we find *k* = 5 to be working well on the 10 tasks setup of Landmarks v2/Celeb-A, as well as on the 50 tasks setup (Sec. 4.2) of Custom Concept.
>
> 5. **W.5: Editorials:** Thank you for pointing this out. Our notation assumes each concept to be a pair of a class label and its respective input images. Therefore, we refer to concepts and class labels interchangeably throughout the manuscript (we will mention this explicitly in the final manuscript).  On this note, we have updated line 103 to read “ .. a text prompt *encompassing* **$c$**” rather than “ .. a text prompt **$c$**” (highlighted in blue). We have also updated line 245 as “.. the one-hot ground truth *for* concept **$c_n$** rather than “.. the one-hot ground truth  **$c_n$**”. Lastly, per your suggestion, we have changed the superscripts for  **$c$** to be subscripts instead.
>
> 6. **W.6: Equation 5 fix:** Thank you for the suggestion, we have updated Eq. 5 accordingly.
>
> 7. **W.7: On the use of 'secret' in the title:** The keyword ‘secret’ emphasizes that conditional diffusion models, while primarily designed for generative tasks, inherently encode class-conditional representations that can be used to estimate the likelihood of data belonging to different classes. Consequently, we propose to use this classification score of text-to-image diffusion models as a regularization basis for continual concept learning. We also note that previous works have used the word 'secret' to emphasize on the classification ability of diffusion models [1].
>
> 8. **Q.1: On the limitation of C-LoRA:** Thank you for your comment. We believe that there has been a **misunderstanding** regarding the behavior of L_forget in C-LoRA. You are *partly* correct in that a straightforward interpretation of L_forget being close to zero indeed appears like no forgetting is happening. However, our limitation discussion in Sec 3.1 further pinpoints a **degenerate case** of C-LoRA, where L_forget is close to zero because the learned weights for LoRA are pushed toward zero (see Fig.1b). This implies that there is no learning happening, in addition to the model having forgotten the previous concept (since most LoRA matrix spots have been edited to approach near-zero values). This is especially evident for task 2 training curves where L_forget decreases throughout training and thus edits most of the learned task-1 LoRA weight spots while pushing these toward zero.
>
> 9. **Q.2: Difference between Sec. 3.1 results and the proposed method:** We believe that this is another case of misunderstanding related to the misinterpretation of C-LoRA's L_forget loss in #8. There is indeed a difference between C-LoRA results and that of ours in the sense that we do not leverage the L_forget loss (given its flaw) shown in Fig. 1 as an optimization objective in our framework. We therefore do not show these curves for our method.
>
> 10. **Q.3: How is $V_n$ obtained and used in the training?** Thank you for your comment. In line 172, we *also* mention that we follow the setup of Custom Diffusion [1] which in turn leverages the setup of Textual Inversion [2] to learn a new textual token embedding **$V_n$** per concept. Since this has been a standard setup for finetuning diffusion models [2,3,4], we do not dive into much details about how **$V_n$** is obtained in the paper. However, following your comment, we have now briefed on this **clearly** in Appendix A (highlighted in blue).
>
>
> **References:**
>
> [1] Li *et al.* "Your Diffusion Model  is Secretly a Zero-Shot Classifier”.  ICCV 2023.
>
> [2] Kumari *et al.* "Multi-Concept Customization of Text-to-Image Diffusion". CVPR 2023.
>
> [3] Gal *et al.* "An Image is Worth One Word: Personalizing Text-to-Image Generation using Textual Inversion".
>
> [4] Smith *et al.* "Continual diffusion: Continual customization of text-to-image diffusion with C-LoRA". TMLR 2024.

---

> ### Author Response · Authors · 2024-11-17
>
> 11. **Q.4: Specific examples of $c_0$, $c_n$, and $V_n$:** We mention the details for **$c_0$** being an abstract pretrained concept “person” in our implementation details in App. H.1. The examples for *n*-th task concept  **$c_n$** are comprehensively listed in App. D, where we provide the concept names for all our datasets. Finally, Sec. 3 *“How do we structure our CL framework for DC scores?”* mentions that  **$V_n$**  is the word vector for the *n*-th task concept **$c_n$** following the Custom Diffusion paper.

---

### Author Response · Authors · 2024-12-01
**Global response**

Dear reviewers,

Thank you for all your comments and constructive suggestions on our manuscript. Your reviews have helped improve our manuscript a lot.

We have been particularly encouraged by your positive response towards several key aspects of our paper including the sound technical contribution, the sufficient experimental results, and the well-addressed limitation of previous work.

We thus express our gratitude for your efforts in helpful reviewing especially given that this can be a potentially overwhelming period of reviewing / responding while preparing your own research.

Thank you again for your effort!

---

### Meta-Review · Area_Chair_T7th · 2024-12-23

**Metareview:**

This paper proposes a framework leveraging diffusion classifier scores for continual personalization of text-to-image diffusion models. The reviewers recognized the paper's technical depth, well-documented results, and relevance to addressing catastrophic forgetting in continual learning setups. However, concerns were raised about the novelty of the proposed method, clarity in presenting its objectives, and computational complexity compared to simpler alternatives like C-LoRA. The authors addressed these concerns through extensive rebuttals, providing additional experiments and detailed clarifications, which satisfied some reviewers. With two reviewers leaning towards weak acceptance and one remaining neutral, the consensus leans towards acceptance as a poster presentation.

**Additional Comments On Reviewer Discussion:**

During the discussion phase, reviewers focused on the method's novelty, computational efficiency, and clarity. Concerns were raised regarding the interplay between DSC and EWC, the computational overhead of the proposed framework, and the necessity of testing on newer stable diffusion versions. The authors clarified the complementary roles of DSC and EWC through additional experiments and included performance benchmarks on Stable Diffusion v2.0. They also demonstrated significant efficiency improvements under certain task setups. Despite resolving most concerns, the novelty and added complexity of the framework compared to simpler baselines like C-LoRA remained partially addressed, influencing reviewer opinions. Ultimately, the reviewers reached a positive consensus, citing the paper's sound methodology and relevance to the field.

---

### Decision · Program_Chairs · 2025-01-22

Accept (Poster)